# Acute deletion of TET enzymes results in aneuploidy in mouse embryonic stem cells through decreased expression of *Khdc3*

Romain O. Georges[1,6], Hugo Sepulveda [1,6], J. Carlos Angel [1], Eric Johnson [1], Susan Palomino[1], Roberta B. Nowak[1], Arshad Desai[2,3], Isaac F. López-Moyado[1,4] & Anjana Rao [1,4,5] ✉

TET (Ten-Eleven Translocation) dioxygenases effect DNA demethylation through successive oxidation of the methyl group of 5-methylcytosine (5mC) in DNA. In humans and in mouse models, TET loss-of-function has been linked to DNA damage, genome instability and oncogenesis. Here we show that acute deletion of all three *Tet* genes, after brief exposure of triple-floxed, Cre-ERT2-expressing mouse embryonic stem cells (mESC) to 4-hydroxytamoxifen, results in chromosome mis-segregation and aneuploidy; moreover, embryos lacking all three TET proteins showed striking variation in blastomere numbers and nuclear morphology at the 8-cell stage. Transcriptional profiling revealed that mRNA encoding a KH-domain protein, *Khdc3* (*Filia*), was downregulated in triple TET-deficient mESC, concomitantly with increased methylation of CpG dinucleotides in the vicinity of the *Khdc3* gene. Restoring KHDC3 levels in triple Tet-deficient mESC prevented aneuploidy. Thus, TET proteins regulate *Khdc3* gene expression, and TET deficiency results in mitotic infidelity and genome instability in mESC at least partly through decreased expression of KHDC3.

The three members of the Ten-Eleven Translocation (TET) enzyme family (TET1, TET2, and TET3) share a highly conserved C-terminal catalytic domain through which they alter the DNA cytosine modification status of the genome. TET enzymes are Fe(II)- and 2-oxoglutarate-dependent dioxygenases that sequentially oxidize the methyl group of 5-methylcytosine (5mC) to 5-hydroxymethylcytosine (5hmC), 5-formylcytosine (5fC) and 5-carboxylcytosine (5caC) in DNA[1–4]. All three oxidized methylcytosines are intermediates in DNA demethylation[1,5–7]. DNA methylation and hydroxymethylation are known to influence many biological processes, such as embryonic development, cell differentiation, cell reprogramming, epigenetic regulation, stem cell function, and oncogenesis[8–13].

In humans, *TET2* loss-of-function mutations are frequent in clonal hematopoiesis (a premalignant syndrome)[13–15] and a variety of haematopoietic malignancies, including diffuse large B cell lymphoma (DLBCL)[16–18], peripheral T cell lymphoma (PTCL) and angioimmunoblastic T cell lymphoma (AITL)[19–23], and myeloid malignancies including myelodysplastic syndrome (MDS), myeloproliferative neoplasms (MPN), chronic myelomonocytic leukemia (CMML) and acute myeloid leukemia (AML)[24–30]. These conditions are most frequent in older individuals. Similarly, in mouse models, germline deletion of the *Tet1* or *Tet2* genes is associated with B cell and myeloid malignancies respectively, and in each case develops with very long latency[31–34]. TET gene mutations are less frequent in non-hematopoietic malignancies,

[1]Division of Signaling and Gene Expression, La Jolla Institute for Immunology, 9420 Athena Circle, La Jolla, CA 92037, USA. [2]Section of Cell and Developmental Biology, Division of Biological Sciences, University of California, San Diego, La Jolla, CA 92093, USA. [3]Department of Cellular and Molecular Medicine, University of California, San Diego, La Jolla, CA 92093, USA. [4]Sanford Consortium for Regenerative Medicine, 2880 Torrey Pines Scenic Drive, La Jolla, CA 92037, USA. [5]Department of Pharmacology and Moores Cancer Center, University of California, San Diego; 9500 Gilman Drive, La Jolla, CA 92093, USA. [6]These authors contributed equally: Romain O. Georges, Hugo Sepulveda. ✉e-mail: arao@lji.org

but hypoxia, deregulation of TET mRNA/protein levels, and a variety of metabolic derangements that downregulate TET catalytic activity can induce profound TET loss-of-function and have been associated with a broad array of both blood and solid cancers[28–30].

Mutations in two or more TET proteins are rarely observed in humans. However, mice engineered to lack two or more TET proteins succumb to cancer considerably more rapidly than mice deficient in a single TET protein, allowing us to investigate the underlying mechanisms in well-controlled systems in vivo. Whereas mice deficient in *Tet1* or *Tet3* alone can survive past birth and mice deficient in *Tet2* alone are viable and fertile, mice doubly deficient in more than one TET protein show diverse and, in some cases, poorly penetrant phenotypes of embryonic lethality[35–37]. Similarly, whereas mice deficient in *Tet2* alone develop myeloid and occasionally lymphoid malignancies with long latency, succumbing by ~400 days[32–34], mice with double deficiency of *Tet2* and *Tet3* in T, B, and myeloid cells develop aggressive cancers. Specifically, double deficiency of *Tet2* and *Tet3* in T cells, induced developmentally with *CD4Cre*, results in rapid, antigen-driven development of a T cell lymphoma involving a normally minor "iNKT" cell subpopulation, that is 100% penetrant and fatal by 5-8 weeks after birth[38,39]. Likewise, double deficiency of *Tet2* and *Tet3*, induced in adult *Mx1Cre* or *CreERT2* mice by injection with polyI:polyC or tamoxifen, respectively, results in fully penetrant development of an aggressive, inducible myeloid leukemia that is fatal by 4-5 weeks[3]. Finally, double deficiency of *Tet2* and *Tet3*, induced in mature B cells with *CD19Cre*, results in a DLBCL-like germinal center B cell lymphoma that is fatal in all mice between 10 and 20 weeks[40]. In each case, the expanded TET-deficient cells show an increased incidence of DNA double-strand breaks (DSBs), as judged by γH2AX levels[3,38–40]; a similar increase in DNA DSBs has been noted in B cell lymphomas that develop in old TET1-deficient mice[8]. Moreover, malignant TET2/3-deficient myeloid cells show a marked delay in the resolution of γH2AX after X-irradiation[3]. Together, the data imply that TET function is necessary for efficient DNA damage repair and hence for genome stability.

Most studies of genome instability in TET-deficient cells have been performed on somatic cells or in stem cell models that have been maintained in culture for variable amounts of time[41,42], enabling the development of compensatory mechanisms that could obscure pathways directly regulated by TET enzymes. In this study, we investigated the effects of acute, profound TET loss-of-function on chromosome segregation and stability in mouse embryonic stem cells (mESC). We find that acute induction of TET deficiency in mESC is swiftly followed by chromosome mis-segregation, which leads to an increase in the frequency of cells with aneuploid karyotypes. Through transcriptomic analysis of *Tet* triple-floxed (*Tet Tfl*) mESC in which *Tet1, Tet2,* and *Tet3* genes were acutely deleted by treatment with 4-hydroxytamoxifen (4-OHT), we identified three target genes, *Khdc3* (*Filia*) and to a lesser extent *Khdc2* (*Floped/Ooep*) and *Nlrp4f*, whose expression was downregulated in TET-deficient mESC. Triple TET deficiency resulted in increased DNA methylation in the vicinity of the *Khdc3* gene, and to a lesser extent the *Khdc2* gene, in mESC, and ectopic expression of KHDC3 in *Tet Tfl* mESC prevented the development of aneuploidy after *Tet* genes deletion. Our data imply that TET proteins, which have been tied to DNA damage responses in mESC as well as other cell types, prevent mitotic abnormalities and maintain chromosome stability in mESC at least partly by maintaining the proper expression of KHDC3.

## Results

### Generation of mouse embryonic stem cell lines for conditional deletion of TET enzymes

To assess the early effects of complete TET loss-of-function (LOF) on chromosome segregation, we generated isogenic inducible mouse embryonic stem cell (mESC) lines in which all three *Tet* genes (*Tet1, Tet2,* and *Tet3*) could be simultaneously deleted following activation of the tamoxifen-inducible Cre-ERT2 fusion protein (Fig. 1A). The mESC

were derived from *Tet Tfl* (*Tet* triple floxed) *Tet1*[fl/fl]; *Tet2*[fl/fl]; *Tet3*[fl/fl]; *Cre-ERT2*; *Rosa26-H2B-LSL-Egfp* mice generated in-house; the *H2b-Egfp reporter* was included for cell tracing and chromosome segregation analysis; as controls, we used identical mESC lines derived from *Cre-ERT2*; *Rosa26-H2B-LSL-Egfp* mice that lacked the floxed *Tet* alleles (here referred to as control, *Ctrl*; for details, see *Methods*). Both *Ctrl* and *Tet Tfl* mESC were cultured in 2i media to maintain them in the ground state of pluripotency and minimize spontaneous differentiation before and after *Tet1/2/3* deletion.

We exposed the *Ctrl* and *Tet Tfl* mESC lines to 4-hydroxytamoxifen (4-OHT) for 2.5 days to induce translocation of the Cre-ERT2 fusion protein into the nucleus and drive recombination of the *loxP* sites flanking *Tet* exons and the LSL cassette [see *Methods*]. The resulting mESC lines−*Ctrl* and *Tet iTKO* (*iTKO*, inducible triple knockout) were then grown for 4 additional days in 2i media without 4-OHT to prevent secondary effects from 4-OHT and minimize the nuclear presence of Cre-ERT2 that might cause Cre-induced genomic instability. *LoxP* recombination occurred in over 90% of the mESCs, assessed at either 3.5 or 6.5 days post-4-OHT treatment as judged by expression of the H2B-EGFP reporter (Supplementary Fig. 1). Reporter expression in *Tet iTKO* mESCs was accompanied by deletion of the *Tet* floxed alleles: RNA-Seq showed no reads in the floxed exons of *Tet1* and *Tet2* at day 6.5 (Fig. 1B), a result confirmed by qRT-PCR. Corroborating previous studies, *Tet3* transcripts were present only at very low levels in *Ctrl* mESC (Supplementary Fig. 2A) but reads from the floxed *Tet3* exons were also lost after exposure to 4-OHT (Supplementary Fig. 2B). TET1 and TET2 protein levels were undetectable following 4-OHT exposure (Fig. 1C and Supplementary Fig. 2C). Consistent with *Tet1/2/3* gene deletion, *Tet iTKO* mESC showed a rapid decrease of 5-hydroxymethylcytosine (5hmC) assessed by flow cytometry (Fig. 1D, E).

### Acute deletion of *Tet* genes in mESC results in increased chromosome mis-segregation

We previously documented a paradoxical decrease in heterochromatic DNA methylation in all TET-deficient cell types analyzed, including mESC[39]. Given that pericentromeric and centromeric regions reside in heterochromatin[43,44], we asked whether acute TET loss-of-function affected chromosome segregation during mESC mitosis. Cre-mediated excision after 4-OHT treatment resulted in deletion of the *LSL* (*loxP*-STOP-*loxP*) cassette located between *H2b-Egfp* and the *Gt(ROSA)26Sor* promoter and consequent expression of the chromatin-associated H2B-EGFP fusion protein, allowing real-time live-cell imaging of segregating chromosomes during mESC mitosis over a 48-hour period between days 4 and 6 after exposure to 4-OHT. Even as early as 4 days after exposure to 4-OHT, we observed chromosome mis-segregation (lagging chromosomes, micronuclei) in *Tet iTKO* mESC at twice the rate observed in *Ctrl* mESC (Fig. 2A, B). These data suggested a direct or indirect role for TET enzymes in protecting accurate chromosome segregation in mESC, a process crucial for equal partitioning of the genome into daughter cells during mitosis.

### Acute deletion of *Tet* genes in mESC results in increased aneuploidy

Since chromosome segregation was clearly affected after acute disruption of all three *Tet* genes, we asked whether *Tet iTKO* mESC displayed an increase in aneuploid karyotypes, a hallmark of genome instability that usually impairs cell differentiation and increases neoplastic potential. Initial low-coverage whole-genome sequencing (WGS) experiments did not detect aneuploidy in bulk populations of *Ctrl* and *Tet iTKO* mESC after acute (6.5 day) exposure to 4-OHT (Supplementary Fig. 3), potentially because random gains and losses of chromosomes in individual cells were obscured in the bulk population, as expected for non-transformed cells[45–47]. We therefore asked whether analysis of single cells would uncover evidence of aneuploidy.

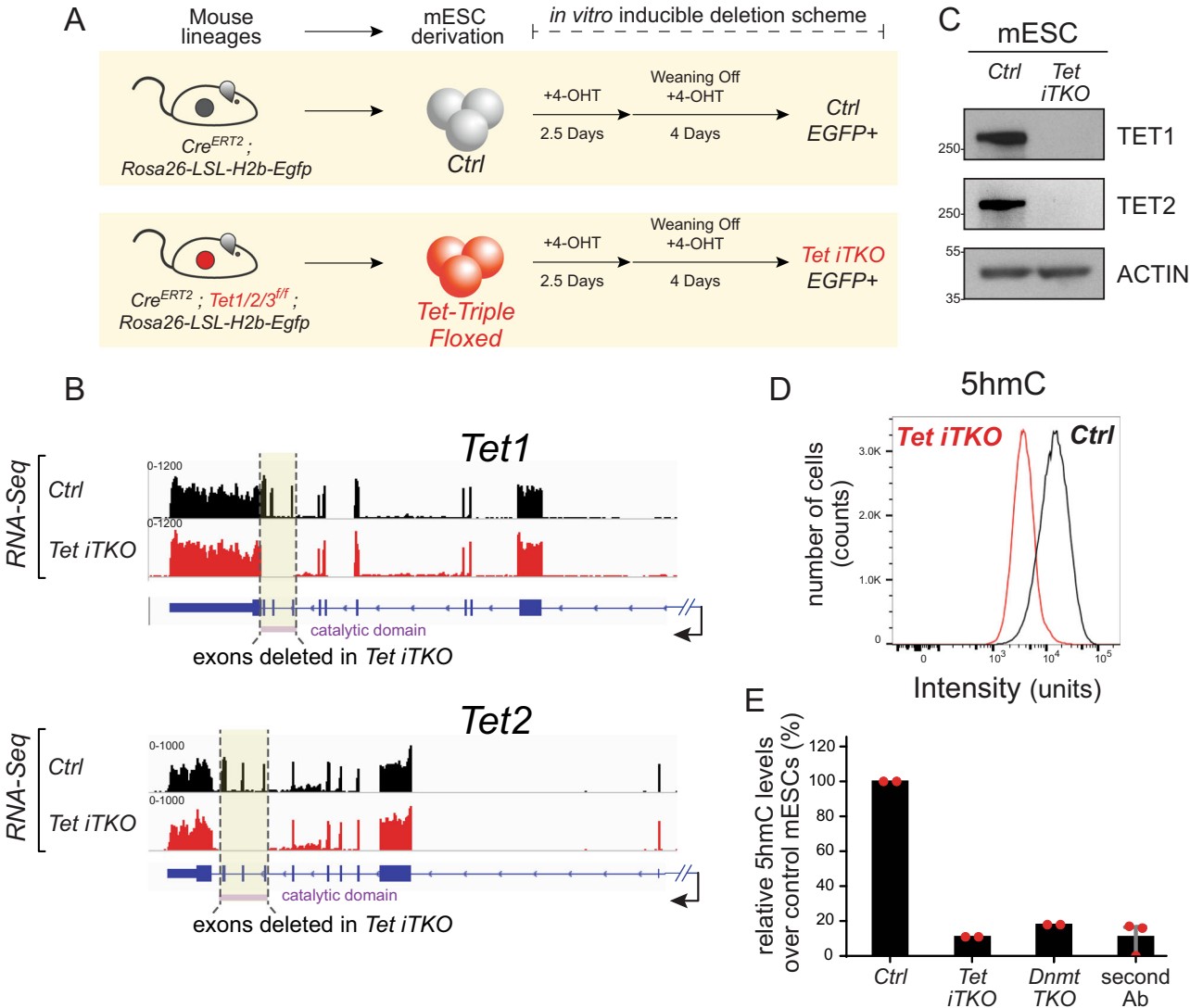

**Fig. 1 | Generation of a tamoxifen-inducible *Tet1/2/3* triple-deletion system in mESC. A** Mouse strains expressing the CreERT2 recombinase for inducible deletion were established in house and used to derive mESC populations that were grown under 2i conditions. Genotypes were Control (*Ctrl*): *CreERT2; Rosa26-LSL-H2b-Egfp*; *Tet triple floxed (Tet Tfl)*: *Tet1/2/3fl/fl; CreERT2; Rosa26-LSL-H2b-Egfp*. Acute deletion of all three *Tet* genes to yield *Tet1/2/3* inducible knockout (*Tet iTKO*) mESC, with concomitant expression of the H2B-eGFP reporter, was accomplished by culturing the cells for 2.5 days in 4-OHT, followed by 4-OHT removal and culture for an additional 4 days before analysis. **B** Genome browser views of *Tet1* and *Tet2* transcripts generated by RNA-Seq analysis of *Ctrl* (black tracks) and *Tet iTKO* (red tracks) mESC. Note that the deleted exons encode the TET catalytic domains (purple bar, *bottom*). **C** Loss of TET1 and TET2 protein expression confirmed by western blot on day 6.5 after exposure of *Ctrl* and *Tet iTKO* mESC to 4-OHT. ACTIN was used as a loading control. **D** Flow cytometric evaluation of 5hmC levels in *Ctrl* and *Tet iTKO* mESC. **E** Quantification of the loss of 5hmC in *Tet iTKO* mESC in duplicate experiments, relative to *Ctrl* and *Dnmt1, Dnmt3a, Dnmt3b triple knockout* (*Dnmt TKO*) mESC ($n = 2$). Signals detected from mESC stained using only the secondary antibody were also included as negative controls ($n = 3$). Bar graphs represent the mean +/− standard error. *Tet iTKO* mESC were gated on H2B-EGFP + / CD90.2-/SSEA1+ /CD326 + cells (see Methods and Supplementary Fig. 2).

Initial attempts at single-cell WGS[48] were not successful, and we therefore turned to performing metaphase spreads on *Ctrl* and *Tet iTKO* mESC (Fig. 2C–E). Metaphase spreads prepared 6.5 days after exposure of *Ctrl* and *Tet iTKO* mESC to 4-OHT revealed a 2- to 3-fold increase in aneuploid karyotypes in three different lines of *Tet iTKO* mESC compared to *Ctrl* mESC (~46-61% versus ~20% aneuploid cells) (Fig. 2C, D). In *Tet triple-floxed* (*Tet Tfl*) mESC not treated with tamoxifen ("untreated" cells in Fig. 2C), the basal frequency of aneuploidy was ~21%. Together, these results confirmed that the chromosome mis-segregation observed at the single-cell level in *Tet iTKO* mESC (Fig. 2A, B) resulted in the rapid appearance of aneuploid karyotypes in individual TET-deficient cells.

There are at least two reasons why aneuploidy that is obvious at the single-cell level might not have been observed in the bulk population: either there is no selective advantage for particular chromosome gains or losses in the bulk population or cells with chromosome instability are eliminated due to impaired fitness. To distinguish these possibilities, we sorted single EGFP+ cells at day 6.5 after 4-OHT treatment and expanded them to obtain clonal populations of long-term deleted *Tet iTKO* mESC (> 80 cell divisions). These long-term deleted clonal populations of *Tet iTKO* mESC showed a similar high frequency of aneuploid karyotypes as the acutely (6.5 day) deleted *Tet iTKO* mESC (Supplementary Fig. 4), confirming that aneuploidy per se is not deleterious to mESC proliferation and survival.

To establish whether the chromosome mis-segregation defect caused by triple *Tet* gene deletion in cultured mESC also occurred in vivo, we performed immunofluorescence staining on early (8-cell stage) embryos (E2.5) in which all 3 TET enzymes were deleted by crossing male *Tet1/2/3* triple floxed (*Tet Tfl*) *Stra8-Cre* mice to female *Tet Tfl Zp3-Cre* mice (see Methods). Eight-cell *Tet TKO* embryos

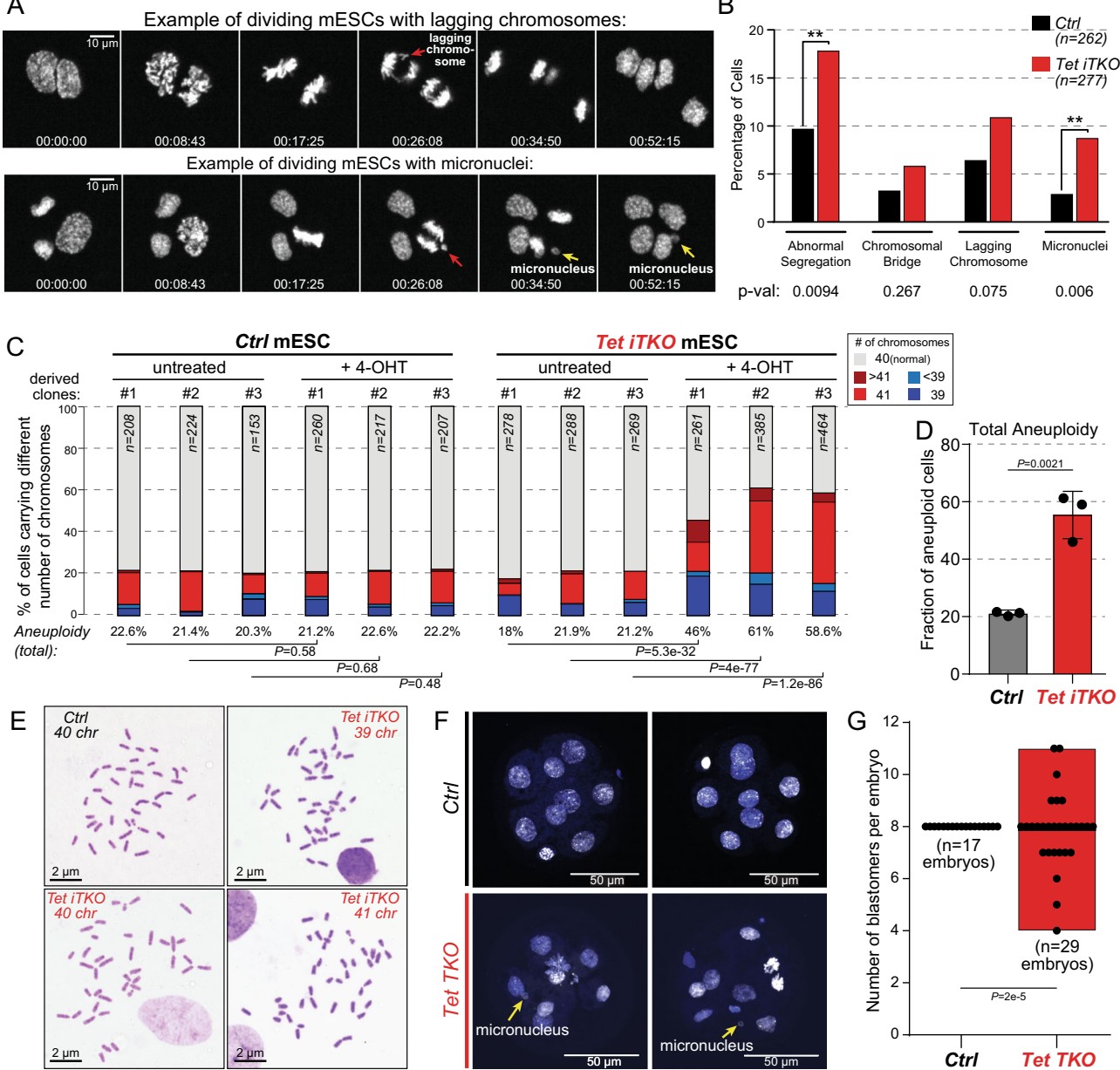

**Fig. 2 | Appearance of aneuploidies and chromosome segregation defects in triple TET-deficient embryos and mESC. A** Time-lapse microscopy of single cells progressing through mitosis. Top panel, image of a lagging chromosome (red arrow) during anaphase. *Bottom* panel, images of a lagging chromosome (red arrows) engendering micronuclei (yellow arrows) after completion of mitosis. Relative times are shown as time stamps at the bottom of each panel representing the time elapsed between each snapshot for two independent movies. **B** Quantification of chromosome mis-segregation data after acute *Tet1/2/3* gene deletion in mESC. Live-cell imaging was performed over 48 h (days 4–6 after addition of 4-OHT), and abnormal chromosome segregation and the presence of micronuclei were quantified. Cochran-Mantel-Haenszel test was performed to calculate p-values from two independent experiments (**p < 0.01). **C** Ploidy analysis calculated from metaphase spreads of *Ctrl* versus acutely-deleted *Tet iTKO* mESC, 6.5 days after exposure to 4-OHT. Percentages of euploid (gray, 40 chromosomes) and aneuploid mESC are represented as bar graphs showing hyperploid cells (>40 chromosomes) in shades of red and hypoploid cells (<39 chromosomes) in shades of blue. Chi-square test was performed, and the calculated p-values are included for

each indicated comparison. **D** Bar graph summarizing the data from cells treated with 4-OHT in **C**. Bar graphs represent the mean +/− standard deviation. Statistical differences between the averages of aneuploidies observed in three *Ctrl* mESC (*Ctrl* mESC + 4-OHT) and three *Tet iTKO* mESC lines (*Tet iTKO* mESC + 4-OHT) were calculated using two-tail unpaired *t*-test. **E** Representative images of metaphase spreads from *Ctrl* and *Tet iTKO* mESC (*n* = 3). **F, G** Representative images (**F**) and quantification (**G**) of the number of blastomeres in *Ctrl* and *Tet TKO* E2.5 embryos obtained from time-mated *Tet1fl/fl; Tet2fl/fl; Tet3fl/fl* breeders for *Ctrl* embryos, or breeders harboring gametes deleted for all 3 TET enzymes for *Tet TKO* embryos (male breeder genotype: *Stra8-Cre; Tet1fl/fl; Tet2fl/fl; Tet3fl/fl* x female breeder genotype: *Zp3-Cre; Tet1fl/fl; Tet2fl/fl; Tet3fl/fl*. *Ctrl* E2.5 embryos (*n* = 17) showed precisely 8 cells in each embryo whereas 15 of 29 *Tet TKO* E2.5 embryos (*n* = 29) displayed abnormal numbers of blastomeres ranging from 4 to 11. Floating bars from *Ctrl* and *Tet TKO* embryos show the distribution of the number of blastomeres for each condition (using min and max numbers as limits). Statistical differences of the frequency of 8-cell stage embryos with an uneven number of blastomeres between *Ctrl* and *Tet TKO* E2.5 embryos were performed using Chi-square test.

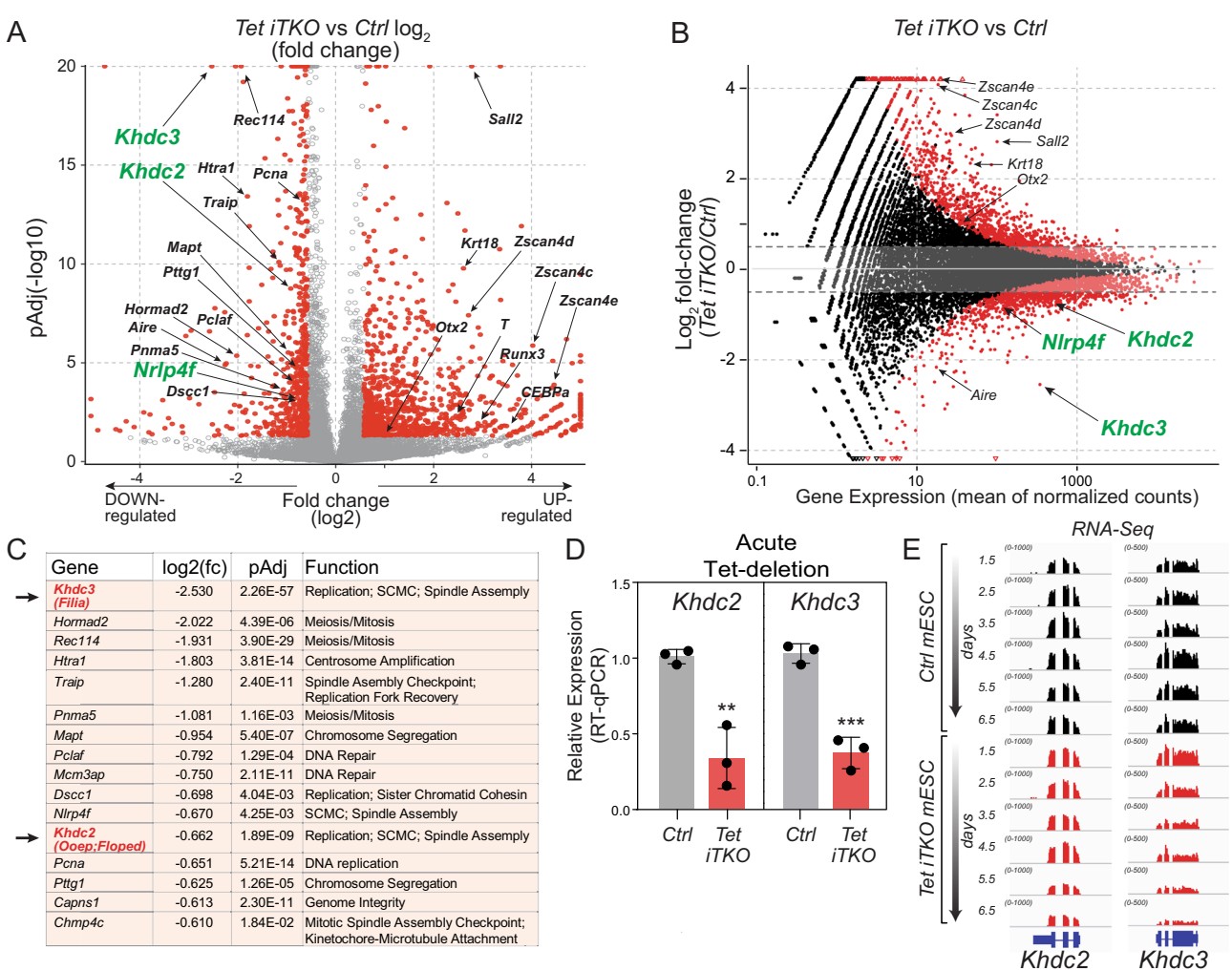

**Fig. 3 | Acute *Tet1/2/3* depletion induces altered expression of many genes, including downregulation of mRNAs encoding the SCMC subunits KHDC2 and KHDC3. A** Volcano plot of differentially expressed genes (DEGs) in acutely-deleted *Tet iTKO* relative to *Ctrl* mESC. The data are from Smart-Seq2 libraries prepared from sorted H2B-EGFP + /CD90.2-/SSEA1 + /CD326 + *Ctrl* or *Tet iTKO* mESC at day 6.5 post-4-OHT exposure. Each point represents an individual transcript. Red dots represent transcripts exhibiting log2 fold change (log2FC) of +/−1 and a pAdj value <0.05 (calculated by adjusting the p-value with the false discovery rate [FDR]). Arrows pinpoint key DEGs. *Khdc2* (*Ooep*), *Khdc3* (*Filia*) and *Nlrp4f* mRNAs, which are downregulated in *Tet iTKO* compared to *Ctrl* mESC and encode core (KHDC2) and more peripheral (KHDC3, NLRP4F) components of the subcortical maternal complex (SCMC), are highlighted. Data are from three independent experiments. **B** MA plot of the same RNA-Seq data, again pinpointing key DEGs and highlighting *Khdc2*, *Khdc3* and *Nlrp4f*. **C** List of downregulated genes whose products are components of, or function in, mitosis/ meiosis, chromosome segregation, DNA replication/ repair, or other aspects of genome integrity. *Khdc2* and *Khdc3* are highlighted. **D** qRT-PCR confirming downregulation of *Khdc2* and *Khdc3* mRNA in *Ctrl* and *Tet iTKO* mESC 6.5 days after 4-OHT exposure (n = 3). Bar graphs represent the mean +/− standard error. The *p*-value was calculated using two-tailed student's *t*-test (**p < 0.01; ***p < 0.001). **E** Time course of downregulation of *Khdc2* and *Khdc3* mRNA in *Ctrl* and *Tet iTKO* mESC (data are from a single experiment).

resulting from this breeding scheme had nuclei that were strikingly heterogeneous in morphology and size, with variable numbers of blastomeres, micronuclei, and blastomere fragmentation presumably reflecting chromosome mis-segregation that would lead to aneuploidy (Fig. 2F, G and Supplementary Fig. 5). The definition of aneuploidy—an abnormal number of chromosomes—encompasses both gains and losses of chromosomes. "8-cell stage" embryos with a reduced number of blastomeres could reflect apoptotic death of a subset of blastomeres or fragmented blastomeres, as observed in some embryos, whereas the presence of an increased number of unevenly sized blastomeres might result from a slight acceleration of the cell cycle in some *Tet* TKO cells or embryos, resulting in premature blastomere division and hence increased blastomere numbers. Indeed, the failure of triple TET-deficient embryos to develop beyond gastrulation[49,50] may at least partially reflect the developmental arrest caused by chromosome mis-segregation during early embryogenesis, as early as the 8-cell stage.

## Acute TET loss-of-function in mESC alters the expression of regulators of chromosome segregation

To identify genes and pathways altered in TET-deficient cells that could affect chromosome segregation, we performed bulk poly(A+) RNA-Seq to analyze the transcriptome of *Tet iTKO* mESC 6.5 days after treatment with 4-OHT (Fig. 3A, B). Pluripotency markers such as *Pou5f1*, *Nanog*, *Essrb*, *c-Myc*, *Klf4*, *Prdm14*, and *Lin28* were expressed at comparable levels in *Ctrl* and *Tet iTKO* mESC (Supplementary Fig. 6A), suggesting that the ground state of pluripotency was not affected by acute *Tet1/2/3* gene deletion. This was confirmed by analyzing the protein levels of OCT4, which remained unaltered after acute *Tet1/2/3* gene deletion (Supplementary Fig. 6B). A small proportion of primed pluripotency-specific and other genes (*Otx2*, *Sall2*, *Krt18*) showed increased expression in *Tet iTKO* mESC (Fig. 3A); the stable and persistent expression of pluripotency markers with co-expression of a small proportion of primed state-specific genes suggests that *Tet iTKO* mESC are for the most part in a metastable naïve state, with perhaps a

minor proportion of *Tet iTKO* mESC progressing towards the primed state. As previously reported[51], acute deletion of *Tet* genes was associated with increased expression of genes in the *Zscan4* cluster, as well as a small number of genes encoding chromatin remodeling factors (Fig. 3A, B).

GO analysis did not point to entire pathways or cellular processes –including genome stability, DNA repair, cell cycle, or apoptosis– that were upregulated in acutely-deleted *Tet iTKO* mESC; however, a significant subset of down-regulated genes encoded important regulators of mitosis, chromosome segregation, spindle assembly, replication, and/or DNA repair (Fig. 3C). Among these acutely-downregulated genes were *Khdc2, Khdc3, Nlrp4f* and *Aire* (Fig. 3A–C); of these, *Khdc3* showed the most striking downregulation. KHDC3 (FILIA, ECAT1) and NLRP4f are peripheral components of the subcortical maternal complex (SCMC), a large multiprotein complex present exclusively in mouse oocytes and pre-implantation embryos[52–56]; MATER (NLRP5), KHDC2 (OOEP, FLOPED) and TLE6 are core components of this complex[53]; and the autoimmune regulator AIRE has been reported to associate with mitotic spindle proteins and control spindle assembly in mESC[57]. We focused on KHDC2 and KHDC3 because they have been reported to form an SCMC-independent complex in mESC[58], and because *Nlrp4f* was only slightly downregulated in acutely deleted *Tet iTKO* mESC (Fig. 3A, B).

*Khdc3* and *Khdc2* mRNAs were downregulated in *Tet iTKO* mESC by day 6.5 after 4-OHT addition (Fig. 3A–E). Time course analysis of the *Tet iTKO* transcriptome showed that expression of both genes gradually decreased with time after *Tet* gene deletion, with *Khdc3* mRNA showing a clear decrease by day 3.5 whereas *Khdc2* mRNA showed a less striking decrease, apparent only by day 5.5 (Fig. 3E). We compared the results from our acute *Tet iTKO* mESC with those from four previously published constitutive *Tet TKO* mESC[51,59–61]; again, only *Khdc3*, not *Khdc2*, was consistently downregulated in these analyzed *Tet TKO* mESC (Supplementary Fig. 7A, B). To obtain insights into the defects noted in *Tet TKO* blastomeres (Fig. 2F, G), we analyzed *Khdc3* expression in *Tet1/3 DKO*[35] and *Tet TKO*[49] embryos; *Khdc3* expression was downregulated in 60% (3/5) of *Tet1/3 DKO* embryos and in all *Tet TKO* embryos compared to normal embryo controls (Supplementary Fig. 7C, D). These observations all point to a prominent role for *Khdc3* deficiency in the aneuploid phenotype of *Tet TKO* mESC. As noted above, long-term deleted clonal populations of *Tet iTKO* mESC continued to display an elevated frequency of aneuploidy (Supplementary Fig. 4A, B); *Khdc3* expression remained downregulated in these cells, whereas *Khdc2* expression returned to normal levels (Supplementary Fig. 4D). Altogether these data suggest that the adverse effects of *Tet* deletion on chromosome segregation might be the consequence of increased DNA methylation correlating with persistent downregulation of *Khdc3* expression.

## DNA methylation of the *Khdc3* promoter in mESC is controlled by crosstalk between TETs and DNMTs

To explore if *Khdc3* expression is controlled by DNA methylation, we analyzed published ChIP-seq, WGBS, and RNA-seq data from selected DNMT- and TET-deficient mESC (Fig. 4). Analysis of published ChIP-seq data from WT mESC[62–65] showed that DNMTs (Fig. 4A, top 3 tracks), especially DNMT3a (2nd track), are enriched in the distal region (−3 kb upstream) of the *Khdc3* transcription start site (TSS). In contrast, TET1 and TET2 occupied regions more proximal to the *Khdc2* and *Khdc3* TSS, as well as *Khdc2* and *Khdc3* gene bodies, in WT mESC (Fig. 4A, 4th and 5th tracks), suggesting mutually exclusive binding patterns of TETs and DNMTs[39,66,67]. Moreover, compared to control mESC, *Tet TKO* mESC showed increased DNA methylation at specific CpGs within the body of the *Khdc3* and to a lesser extent the *Khdc2* gene, as well as in the regions surrounding the two genes[51] (Fig. 4A, bottom 2 tracks), correlating with decreased expression of *Khdc3* and *Khdc2* mRNA (Fig. 4A, 6th and 7th tracks). Conversely, *Khdc3* expression was

increased in *Dnmt TKO* compared to control mESC[59] (Fig. 4B), consistent with the well-established negative correlation (but not necessarily a causal relation) between DNA methylation and gene expression[68–70].

To determine whether one or more specific DNMTs had a major role in controlling methylation of the *Khdc3* locus, we examined published WGBS data from mESC with individual or multiple deficiencies of DNMTs[71,72] (Fig. 4C). *Dnmt3a KO* mESC (Fig. 4C, top 2 tracks) displayed a substantial loss of DNA methylation at specific CpGs of the *Khdc3* promoter compared to WT mESC, whereas *Dnmt3b KO* mESC (middle 2 tracks) did not. In fact, certain CpGs in the *Khdc3* locus showed increased methylation, suggesting re-localization of DNMT3a or DNMT1 to these CpGs in the absence of DNMT3b. *Dnmt3a, 3b DKO* mESC showed complete loss of DNA methylation across the *Khdc3* locus, as did *Dnmt1/3a/b TKO* mESC as expected (bottom 2 tracks). These results point to the de novo DNA methyltransferase DNMT3a as the most important DNMT controlling DNA methylation at the *Khdc3* locus.

We next examined the consequences of sequential deletion of *Tet* and *Dnmt* genes at the *Khdc3* locus[61] (Fig. 4D). The investigators first deleted all three *Tet* genes to generate *Tet TKO* mESC; then sequentially deleted first the *Dnmt3a/b* genes and next the *Dnmt1* gene to yield *Tet/Dnmt3 PKO* mESC (pentuple knockout, 3 *Tets* and both *Dnmt3a, 3b*) and *Tet/Dnmt SKO* mESC (sextuple knockout, all 3 *Tets* and all 3 *Dnmts*) respectively. WGBS analyses of these cell lines showed that double deletion of *Dnmt3a* and *Dnmt3b* prevented DNA hypermethylation at the highlighted regions of the *Khdc3* promoter in *Tet TKO* mESC, although unexpectedly, gene body methylation persisted, presumably reflecting the action of DNMT1 (Fig. 4D, compare top 3 tracks). As expected, mESC with complete deletion of all three *Tet* and *Dnmt* genes in *Tet/Dnmt SKO* almost completely lacked DNA methylation (Fig. 4D, bottom track). Notably, analysis of RNA-seq datasets showed that the decrease of *Khdc3* expression in *Tet TKO* mESC (Fig. 4E, top and middle tracks) is restored to more than WT levels upon further deletion of both *Dnmt3a* and *Dnmt3b* in *Tet/Dnmt3 PKO* mESC (Fig. 4E, bottom track), illustrating an intriguing crosstalk between TET and DNMT proteins[61]. Consistent with the previous findings, DNMT ChIP-seq data showed that DNMT3a is the most enriched DNMT at the *Khdc3* promoter (Fig. 4A) and a previous report indicated that DNMT3a plays a major role in controlling DNA methylation in TET-regulated regions in ESC[73]. Interestingly and in agreement with those previous studies, the study of single *Tet1*- and *Tet2-KO* mESC revealed a predominant role for TET2 in regulating *Khdc3* expression in mESC and maintaining the demethylated state of the *Khdc3* promoter (Supplementary Fig. 8).

## Re-expression of *Khdc3* in *Tet iTKO* mESC reverses the aneuploid phenotype

Previous studies have shown that *Khdc3* promotes genomic stability in ESC, and that disruption of the *Khdc3* gene in mESC results in aneuploidy[74]. To confirm and expand those observations, we used the CRISPR/Cas9 system to target both *Khdc2* and *Khdc3* genes, generating mESC deleted for each gene (Supplementary Fig. 9). Confirming the previous report, we found that disruption of either the *Khdc2* or *Khdc3* resulted in increased aneuploidy. Next, given the downregulation of both *Khdc2* and *Khdc3* in acutely deleted *Tet iTKO* mESC, we asked if re-expression of *Khdc2* or *Khdc3* would prevent the emergence of aneuploidies in *Tet iTKO* mESC. We re-expressed *Khdc2* or *Khdc3* inducible under doxycycline (TET-on) control in *Tet iTKO* and *Ctrl* mESC (Fig. 5 and Supplementary Fig. 10) and performed metaphase spreads as a readout to assess aneuploidy. *Ctrl* and *Tet iTKO* mESC were transduced with a lentivirus harboring the reverse tetracycline-controlled transactivator 3 (*rtTA3*) under the control of the human EF1a promoter; a second step of transduction introduced FLAG epitope-tagged versions of *Khdc2* or *Khdc3* (isoform 1) under control of a doxycycline-inducible

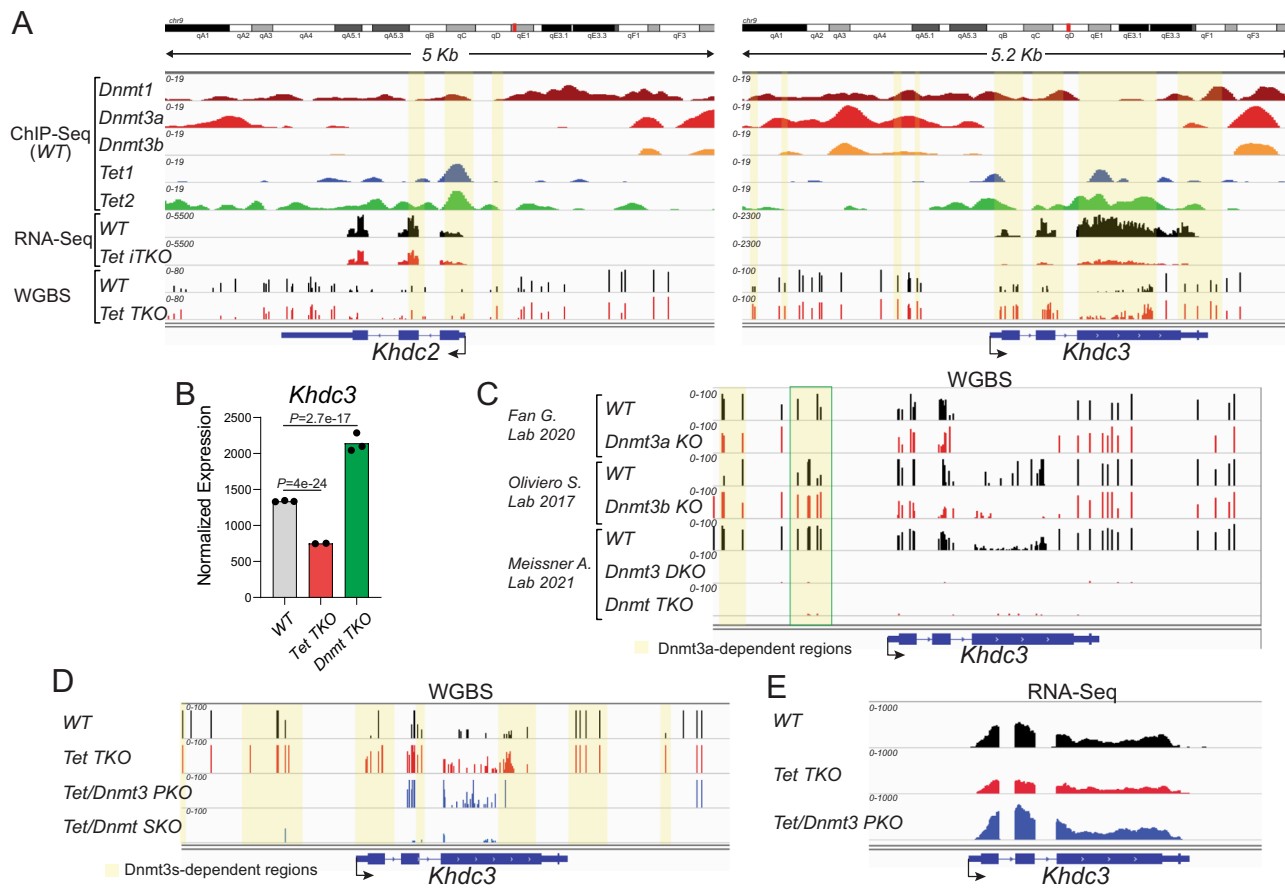

**Fig. 4 | The *Khdc3* promoter is controlled by TETs and DNMTs. A** Genome browser view of the *Khdc2* (left) and *Khdc3* (right) loci in mESC. Top five tracks, ChIP-Seq data for DNMT1 (dark red), DNMT3A (red), DNMT3B (orange), TET1 (blue) and TET2 (green)[62–65]. Middle two tracks, *Khdc2* and *Khdc3* transcripts in *Ctrl* (black tracks) and *Tet iTKO* (red tracks) mESC generated from our Smart-Seq2 analysis. Bottom two tracks, methylation status of CpGs in the *Khdc2* and *Khdc3* loci, based on WGBS data[51]. Notice the increase in methylation at the promoters, gene bodies and nearby CpGs of both the *Khdc2* and especially the *Khdc3* genes. **B** *Khdc3* gene expression from constitutive *Tet TKO* (red) and *Dnmt TKO* (green) compared to *WT* mESC (gray)[61]. Notice the enhanced expression of *Khdc3* in mESC lacking *Dnmts* (green). *P*-adjusted values were calculated respect to *WT* mESC using the Benjamini-Hochberg method to control the False Discovery Rate. **C** DNA methylation maps at the *Khdc3* locus from single *Dnmt3a-* (upper group) or *Dnmt3b KO* (middle group)

compared to their respective controls (*WT*), as well as double *Dnmt3a/b KO* (*Dnmt3 DKO*) or triple *Dnmt1/3a/b KO* (*Dnmt TKO*; bottom group). Notice the loss of DNA methylation at the *Khdc3* promoter in *Dnmt3a KO* (but not *Dnmt3b KO*) mESC and the complete loss of methylation in *Dnmt3 DKO* mESC[71,72]. **D** DNA methylation maps at the *Khdc3*-locus from constitutive *Tet TKO* mESC (*track 2*) that were subsequently modified to remove *Dnmt3a/b* (*Tet/Dnmt-PKO, track 3*) or all three *Dnmts* (*Tet/Dnmt* sextuple knockout *(SKO), track 4*) are compared to control mESC (*WT, track 1*). Notice that removal of *Dnmt3a* and *Dnmt3b* in the *PKO* mESC prevents DNA hypermethylation at the *Khdc3* promoter in *Tet TKO* cells (*track 2*), although unexpectedly, gene body methylation persists[61]. **E** Removal of *Dnmt3a* and *Dnmt3b* results in de-repression of *Khdc3* mRNA expression in *Tet TKO* mESC (blue compared to red; same samples as shown in **D**).

CMV promoter (Fig. 5A). For analysis of aneuploidy by metaphase spreads, we chose *Tet Tfl* and *Ctrl* mESC clones that showed matched expression of KHDC2 and KHDC3 proteins after doxycycline treatment as determined by western blots using an anti-FLAG antibody (Fig. 5B and Supplementary Fig. 10). Given the lack of commercially available antibodies that can detect endogenous KHDC2 and KHDC3, we were unable to assess the relative expression of the introduced versus the endogenous KHDC2 and KHDC3 proteins.

*Khdc2-* and *Khdc3*-expressing *Tet iTKO* and *Ctrl* mESC clones were treated simultaneously with 4-OHT and doxycycline; 4-OHT was removed at day 2.5 whereas doxycycline treatment was maintained until day 6.5, the day of harvest and fixation for preparation of metaphase spreads. In control *Ctrl* mESCs, expression of either *Khdc2* or *Khdc3* did not alter the frequency of aneuploidy, which was maintained at a basal level of around 20% (Fig. 5C, left). However, in *Tet iTKO* mESC that had been reconstituted with *Khdc3*, the frequency of both hyperploidy and aneuploidy decreased from ~60% to ~20%, resembling

the basal frequencies detected in *Ctrl* mESC; in contrast, re-expression of *Khdc2* alone did not reverse the high overall frequency of aneuploidy seen in *Tet iTKO* mESC (Fig. 5C, right). Neither *Khdc2* nor *Khdc3* re-expression affected the expression of the pluripotency marker OCT4 (Supplementary Fig. 10). These data indicate that TET enzymes control normal chromosome segregation during mitosis at least partly by ensuring normal *Khdc3* expression in mESC.

## Discussion

We show here that TET proteins are important for ensuring the fidelity of chromosome segregation in mESC, and that this effect is mediated through downregulation of *Khdc3*. Acute Cre-mediated deletion of *Tet1, Tet2,* and *Tet3* genes results in rapid loss of TET proteins and undetectable levels of 5hmC in mESC by 4–6 days of the addition of 4-OHT to *Tet Tfl* mESC, and the *Tet iTKO* cells display marked mitotic abnormalities, chromosome mis-segregation and aneuploidy. This phenotype is at least partly due to the downregulation of *Khdc3 (Filia,*

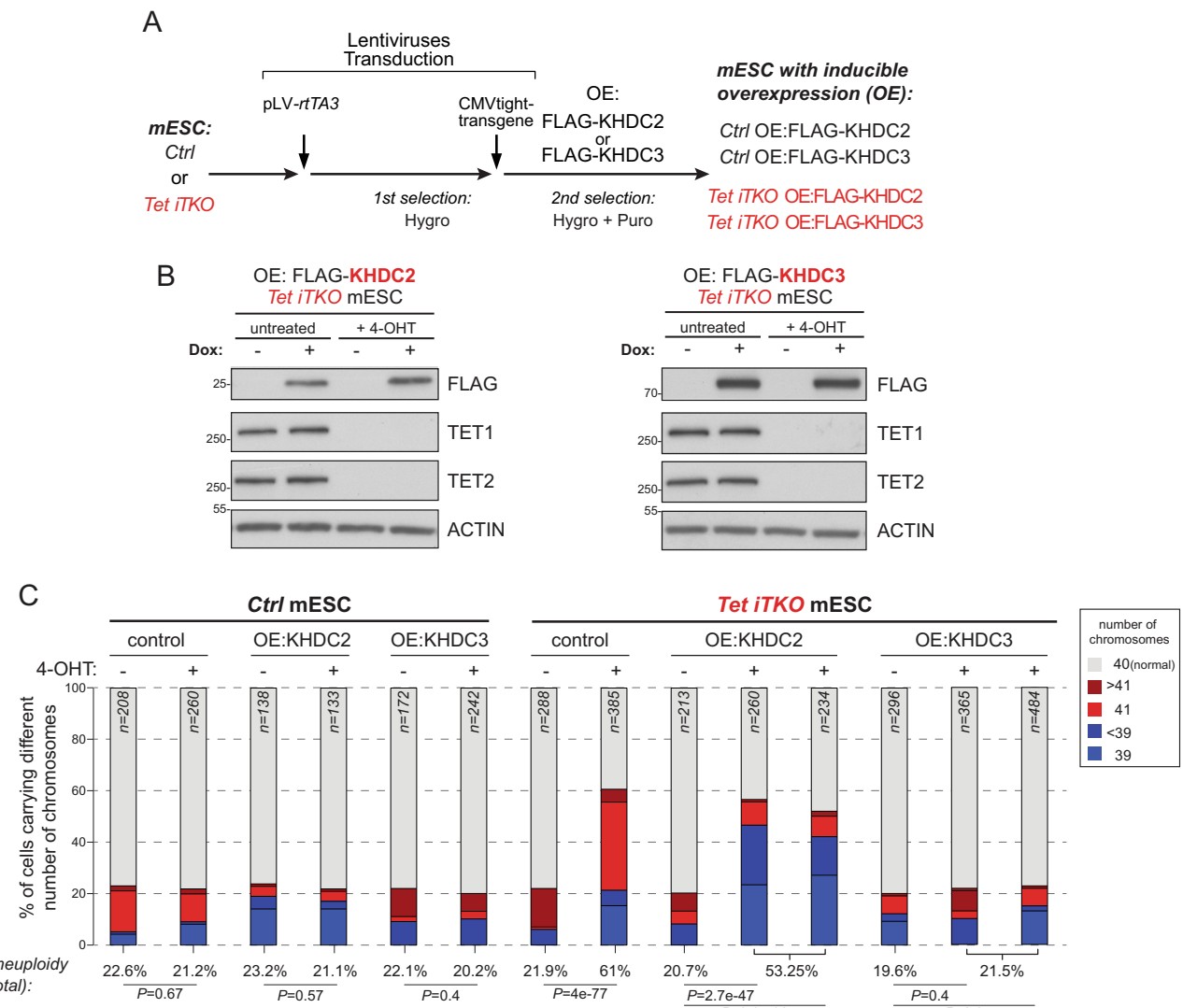

**Fig. 5 | KHDC3 re-expression reverses the aneuploid phenotype of *Tet iTKO* mESC. A** Flowchart of the procedure to generate mESC with doxycycline-inducible overexpression (OE) of KHDC2 or KHDC3 in *Ctrl* (*Cre-ERT2*⁺) and *Tet Tfl Cre-ERT2*⁺ mESC. **B** Western blot analysis of doxycycline-inducible expression of KHDC2 (left) or KHDC3 (right) in *Tet iTKO* mESC (untreated and treated with 4-OHT respectively). KHDC2/3 proteins were detected using anti FLAG-antibodies and the effective removal of TET proteins was confirmed as in Fig. 1C. ACTIN was used as loading control. OCT4 protein expression was unchanged in KHDC2- and KHDC3- expressing *Ctrl* and *Tet iTKO* mESC (Supplementary Fig. 10). **C** Ploidy analysis calculated from metaphase spreads of *Ctrl* versus acutely-deleted *Tet iTKO* mESC after reconstitution with KHDC2 or KHDC3. Percentages of euploid (gray, 40 chromosomes) and aneuploid mESC are represented as bar graphs showing hyperploid cells (>40 chromosomes) in shades of red and hypoploid cells (<39 chromosomes) in shades of blue. There is no effect of KHDC2 or KHDC3 expression on aneuploidy in *Ctrl* mESC (left), but a clear decrease in aneuploidy in KHDC3-expressing (but not KHDC2-expressing) *Tet iTKO* mESC (right).

*Ecat1*), which encodes a KH domain-containing protein originally identified as an interaction partner of the maternal effect gene product NLRP5 (MATER), a core component of the subcortical maternal complex (SCMC)[54]. Expression of *Khdc3* in *Tet Tfl* mESC prevented the rapid appearance of aneuploidy in *Tet iTKO* mESC after 4-OHT exposure, whereas expression of *Khdc2* did not. As expected from the biochemical activities of TET proteins and their involvement in DNA demethylation, downregulation of *Khdc2* and *Khdc3* mRNA was accompanied by increased methylation of CpGs in and around the promoters and gene bodies of both *Khdc2* and *Khdc3* genes, supporting the well-known correlation (not necessarily a causal connection) between increased DNA methylation in the vicinity of genes and decreased gene expression[68–70].

The copy number changes observed in TET-deficient cells are unlikely to be an artefactual consequence of double-strand DNA breaks introduced by Cre recombinase. First, we deliberately minimized the time period of Cre activation by removing 4-OHT on day 2.5,

a strategy that terminates Cre recombinase activity in the nucleus by returning unliganded Cre-ERT2 to the cytoplasm, thus preventing further recombination among real or potential cryptic *loxP* sites in the genome. Second, re-expression of *Khdc3* in *Tet iTKO* mESC−in the same 6.5-day time period as 4-OHT treatment, Cre activation and *Tet* gene deletion−prevented increased aneuploidy, suggesting that aneuploidies triggered only by Cre expression do not constitute a significant proportion of total aneuploidies in *Tet iTKO* mESC. Third, similar phenotypes of chromosome segregation abnormalities and aneuploidy have been reported previously in triple TET-deficient mESC, under conditions in which *Tet* gene deletion was not mediated by Cre[41,75]. Thus, Yang et al.[75] observed chromosomal fusions and telomere shortening after stable retroviral shRNA-mediated depletion of *Tet1* and *Tet2* in mESC, while Kafer et al.[41] observed increased frequencies of mitotic abnormalities in mESC in which all three *Tet* genes were disrupted using CRISPR/Cas9 technology. Since both RNAi and CRISPR/Cas9 strategies require prolonged periods of mESC culture to

select for correctly silenced/ edited cells, we chose to focus on the consequences of acute deletion of *Tet* genes using Cre recombinase in mESC. Our data show clearly that the emergence of genomic instability —manifest as chromosome mis-segregation, copy number changes and aneuploidies—is an early and direct consequence of TET deficiency in mESC, caused in large part by the transcriptional downregulation of *Khdc3* gene expression in TET-deficient cells. As expected from previous studies[41,75], the aneuploid phenotype is maintained when single cells from acutely deleted *Tet iTKO* mESC populations are clonally expanded in long-term culture (> 80 cell divisions).

There are well-known selective pressures that oppose the development of aneuploidy in most normal cell types; in fact, the large majority of chromosome gains and losses are known to impair cell fitness (reviewed in refs. 47, 76). We assume, therefore, that the increased aneuploidy that we observe at a single-cell level (by metaphase spreads) in both acute and long-term *Tet iTKO* mESC cultured in 2i media involve chromosomes whose gains or losses do not seriously hamper mESC proliferation and/or survival, explaining the relative stability of the aneuploid phenotype following *Tet* gene deletion. In this context, we note that random chromosome gains and losses are observed early in cancer development, because selective pressures have not yet come into play; at later stages, specific chromosomal gains and losses may be recurrently observed, presumably because they confer increased fitness by promoting survival and/or expansion of individual clones[45,46] [reviewed in ref. 47].

Among the down-regulated genes encoding regulators of mitosis, chromosome segregation, spindle assembly, DNA replication or DNA repair, *Khdc3* expression was the most strikingly decreased in *Tet iTKO* compared to *Ctrl* mESC. KHDC3 (FILIA, ECAT1) is a peripheral component of the subcortical maternal complex (SCMC), a multiprotein complex that is present exclusively in mouse oocytes, zygotes and pre-implantation embryos[52–56]. The core components of the SCMC – MATER (NLRP5), KHDC2 (OOEP, FLOPED) and TLE6[53] – are required for embryos to progress beyond the 2-cell stage[52,53]. Embryos of female mice deficient in either *Mater/Nlrp5* or *Khdc2* showed a complete block of zygotic development at the two-cell stage;[52,53] in contrast, the phenotype of KHDC3-deficient mouse embryos was much less catastrophic, with a marked delay in early embryo development but not a complete arrest[55]. Female KHDC3-deficient mice showed reduced fertility with smaller litters;[55] similarly, immature human oocytes injected with siRNAs against *Khdc3/Ecat1* showed reduced maturation, decreased fertilization in response to intracytoplasmic sperm injection, and decreased progression to the two-cell stage in vitro[77]. A notable early observation was that cells from KHDC3-deficient mouse embryos frequently displayed micronuclei as well as high rates of aneuploidy compared with normal controls[55], a result we have confirmed here. Moreover, *Khdc3*-null mESC showed increased DNA damage measured by the number of γH2AX-positive foci, as well as delayed kinetics of DNA damage repair[74]. We show a similar phenotype of variable blastomere numbers and abnormal nuclear morphology in triple TET-deficient embryos at the 8-cell stage. Given the well-known association of chromosome mis-segregation with DNA damage[78,79], decreased expression of *Khdc3* may contribute substantially to the increased DNA damage observed in TET-deficient embryos and mESC.

Are there distinct or overlapping roles for *Khdc2* (*Ooep, Floped*) and *Khdc3* (*Filia, Ecat1*) in maintaining genome stability in mESC? In size exclusion chromatography of oocyte lysates, the core SCMC components (NLRP5, TLE6 and KHDC2) co-eluted in a single broad peak with KHDC3, but a substantial amount of KHDC3 also eluted at lower apparent molecular weights[53], suggesting that KHDC3 dissociates readily from the core SCMC complex and thus may act independently or in complexes with non-SCMC proteins. The core SCMC component NLRP5 (MATER) is not expressed in blastocysts or mESC (Supplementary Fig. 11), implying that the SCMC is not present in mESC; rather, a separate complex of KHDC2 and KHDC3 has been detected at nascent (newly replicated) DNA in mESC, together with the replication fork restart proteins BLM and TRIM25[58]. *Khdc3* has been shown to maintain chromosome stability and regulate the DNA damage response in mESC[78]; *Khdc3*-null ES cells showed impairment of the late phase of ATM phosphorylation as well as impaired CHK2 phosphorylation at all time points after exposure to etoposide[74]. Although *Khdc2* and *Khdc3* mRNAs were both rapidly down-regulated in *Tet iTKO* mESC after 4-OHT addition, *Khdc3* expression remained downregulated even after prolonged culture whereas *Khdc2* expression was restored. Thus, *Khdc3* may operate independently of *Khdc2* to prevent chromosome instability in *Tet iTKO* mESC; alternatively, because *Khdc2* mRNA is expressed at higher levels than *Khdc3* mRNA in control mESC and is not as markedly downregulated in either acutely deleted or long-term-cultured *Tet iTKO* mESC, the two proteins may function as a complex in which KHDC3 protein is limiting whereas KHDC2 protein is not. We note that *Khdc3* downregulation cannot account for copy number variations and aneuploidies occurring in hematopoietic malignancies such as DLBCL and PTCL, since *Khdc3* expression is restricted to cells at early stages of embryonic development including mESC (Supplementary Fig. 11).

It is well-established that DNA demethylation caused by DNMT deficiency induces aneuploidies: in *Dnmt1* hypomorphic cells in vivo[80,81], in peripheral T cell lymphomas arising from haematopoietic stem/precursor cells conditionally disrupted for *Dnmt3a*[82,83] and in *Dnmt3b* deficiency mimicking ICF syndrome (immunodeficiency, centromere instability and facial abnormalities) in vivo[84]. To illustrate this point, we analyzed WGBS from *Dnmt* KO ESCs. Supplementary Fig. 12 highlights copy number changes in different types of DNMT-deficient mESC with respect to the corresponding parental lines[71,72], and supports previous data associating DNA hypomethylation with chromosome segregation defects and aneuploidies[80–84]. Thus, an interesting question is whether, in addition to *Khdc3* downregulation, decreased DNA methylation in heterochromatin[39,67,85] contributes to the increased aneuploidies observed in triple TET-deficient mESC. We are currently investigating this possibility.

## Methods

### Mice

Mice were housed in a pathogen-free animal facility at the La Jolla Institute for Immunology and were used according to protocols approved by the Institutional Animal Care and Use Committee (IACUC). To obtain the control inducible mouse strain (*Cre^ERT2*; *Rosa26-H2B-Egfp^LSL* mice, where LSL denotes the *loxP-STOP-loxP* cassette that permits expression of the floxed exons or the reporter gene only after excision by the Cre recombinase), we crossed *B6.Cg-Tg(UBC-cre/ERT2)1Ejb/J* mice (Jackson Laboratory, #008085), harboring the *Cre^ERT2* fusion gene under control of the human ubiquitin C (UBC) promoter, with *Gt(ROSA)26Sor < tm1Ytchn > /J* (Jackson Laboratory, #021847) mice containing a CRE-recombinase inducible dual reporter construct in which *H2b-Egfp* and *GPI-mCherry* transgenes were inserted into the *Gt(ROSA)26Sor* locus downstream of a *loxP*-flanked STOP fragment. The *H2b-Egfp; GPI-mCherry* reporter cassette was chosen for cell tracing and chromosome segregation analysis, but the GPI-Cherry transgene was poorly expressed and was not useful for staining or flow cytometry.

The *Tet-triple floxed* (*Tet Tfl*) mouse strain (*Cre^ERT2*; *Tet1^fl/fl*; *Tet2^fl/fl*; *Tet3^fl/fl*; *Gt(ROSA)26Sor < tm1Ytchn > /J^wt/ki*) harbors *Cre^ERT2* and *H2b-Egfp; GPI-mCherry* transgenes, as well as the floxed alleles for all three *Tet* genes. In *Tet1^fl/fl* and *Tet2^fl/fl* mice, exons 8, 9, and 10 encoding for the catalytic HxD domain of *Tet1* and *Tet2* were floxed (flanked by LoxP sites), whereas exon 2 was targeted in the case of the *Tet3^fl/fl* mouse line. All *Tet floxed* mice were generated from ART B6-3 embryonic stem cells (genetic background: C57BL/6 NTac)[3,32,35,86]. To generate E2.5 (8-cell stage) embryos, *Stra8-Cre^wt/ki*; *Tet1^fl/fl*; *Tet2^fl/fl*; *Tet3^fl/fl* were time-mated with *ZP3-Cre^wt/ki*; *Tet1^fl/fl*; *Tet2^fl/fl*; *Tet3^fl/fl*.

## Derivation of *Tet Tfl* (*Tet* triple-floxed) mESC

*Cre^ERT2 wt/ki*; *Gt(ROSA)26Sor < tm1Ytchn > /J^wt/ki* and *Cre^ERT2 wt/ki*; *Tet1/2/3^fl/fl*; *Gt(ROSA)26Sor < tm1Ytchn > /J^wt/ki* embryos were produced by timed matings of *Cre^ERT2 wt/wt*; *Gt(ROSA)26Sor < tm1Ytchn > /J^wt/wt* with wildtype (*WT*) C57BL/6 J and *Tet1^fl/fl*; *Tet2^fl/fl*; *Tet3^fl/fl* mice. Mouse embryonic stem cells (mESC) were derived from E3.75 blastocysts flushed out from the uterine horn of time-mated females. Blastocysts were collected and hatched on mitomycin C mitotically inactivated MEFs plated 1 day earlier. After attachment, SRES media was replaced every second day until day 6 after attachment of the blastocyst to the feeder MEFs layer. On day 6, the expanded blastocysts were dissociated using TrypLE™ Select Enzyme (10X) (Gibco, A1217702) and dissociated cells were plated on mitotically inactivated MEFs. Cells were grown and SRES media + 2i inhibitors was changed every second day until compact cell colonies with typical ESC colony morphology formed. mESC were passaged 2–3 times before being cryopreserved. All mESC lines were tested routinely and found to be free of any mycoplasma contamination.

## mESC culture and *Tet* gene deletion

mESC were cultured in 2i media over a layer of mitomycin C-treated mouse embryonic fibroblasts (MEFs) to maintain them in their ground state of pluripotency and minimize any potential spontaneous differentiation prior to or after triple *Tet* gene deletion. The culture medium was Serum Replacement ESC medium (SRES medium) composed of KnockOut™ DMEM/F-12 (Gibco, 12660012) supplemented with 15% KnockOut™ Serum Replacement (Gibco, #10828028), GlutaMAX™-I (Gibco, 35050061), 1x MEM non-essential amino acids (Gibco, 11140-035), 100 μM β-mercaptoethanol (Gibco, 21985023), LIF, and 2i (3 μM GSK3 inhibitor CHIR99021 (Abmole Bioscience, M1989) and 1 μM MEK inhibitor PD0325901 (Selleckchem, S1036)).

Acute deletion of all three *Tet* genes to yield *Tet1/2/3* inducible knockout (*Tet iTKO*) mESC, with concomitant expression of the *H2b-Egfp* reporter, was accomplished by culturing the cells for 2.5 days in 4-OHT, followed by 4-OHT removal and culture for an additional 4 days before analysis.

To conditionally delete TET enzymes, mESC plated on mitotically inactivated MEFs were treated with 1 μM 4-hydroxytamoxifen (4-OHT) (Tocris, 3412) for 2 days. After 1.5 additional days of growth in SRES without 4-OHT, H2B-EGFP + mESC, resulting from a successful *Cre^ERT2*-mediated recombination that occurred during 4-OHT treatment, were sorted by FACS (Fluorescence-Activated Cell Sorting) using a FACSAria cell sorter (BD Biosciences). mESC were plated on mitotically inactivated MEFs and allowed to attach for 6 h prior to initiating imaging of the cells for 48 h. *Dnmts TKO* mESC were previously described[87,88].

## Expression of KHDC2 and KHDC3 in *Tet Tfl* mESC

*Khdc2* and *Khdc3* were cloned into a CMVtight Tet-on advanced lentiviral backbone vector (Addgene), which allows the rapid re-expression of any gene of interest after the introduction of doxycycline in the cell media. All constructs were confirmed by Sanger sequencing.

## CRISPR/Cas9 editing of *Khdc2* and *Khdc3* in WT mESC

crRNA-tracrRNA duplex was prepared by annealing equimolar concentrations of Alt-R crRNA and Alt-R tracrRNA reconstituted in 100 uM with NF Duplex Buffer (IDT). RNP complexes were assembled by mixing 10 μg of TrueCut CAS9 protein v2 to the annealed crRNA-tracrRNA duplex at a 1:3 molar ratio and incubated at room temperature for 10 min. Electroporation was carried out with the 4D-Nucleofector Core Unit (LONZA) and using the P3 Primary Cell 4D-Nucleofector X kit (LONZA, V4XP-3032). mESC were gently mixed with RNP complexes and incubated for 2 min. at room temperature before transferring the cells and RNP mix into a nucleofection cuvette strip. A combination of 2 sgRNAs per gene were mixed to edit each gene within two different exons (sgRNA1 against *Khdc2*: CAAAGATGGTGATTTC AGCT using TGTCCAAAGATGGTGATTTCAGCTAGGTTC for context sequence; sgRNA2 against *Khdc2*: TCCTGAATTGGGAACCACCA using GACTTCCTGAATTGGGAACCACCAGGGCCG for context sequence; sgRNA1 against *Khdc3*: GGCTCCCGTGAAGGTCCGCG using AGTTGGC TCCCGTGAAGGTCCGCGAGGCGG for context sequence; sgRNA2 against *Khdc3*: CATTCGACATGGAACCACTT using CAGGGCATTCGA-CATGGAACCACTTGGGCAA for context sequence). As a negative control mESC were nucleofected with a RNP complex assembled with an Universal Non-Targeting Control sgRNA (scrambled sgRNA; AAATGTGAGATCAGAGTAAT). Electroporation was performed using the program CG-104 on the 4D-Nucleofector X-unit. After electroporation, cells were transferred into SRES + 2i media and plated into one well from a six-well dish.

## Lentivirus production and transduction

Lentiviruses were produced by transfecting the Lenti-X™ 293T cell line (Takara, 632180) with the different lentiviral constructs. Lenti-X™ 293T cells were grown in DMEM (Gibco, 11965118) supplemented with 10% Fetal Bovine Serum (Foundation B™ FBS) (GeminiBio, 900-208), GlutaMAX™-I (Gibco, 35050061), 1x MEM non-essential amino acids (Gibco, 11140-035), 100 μM β-mercaptoethanol and 1 mM sodium pyruvate (Sigma-Aldrich, S8636). 48 h after transfection, media containing lentiviruses was collected and lentivirus titers were measured using the Lenti-X qRT-PCR Titration Kit (Takara, 631235). Lentiviruses were added to the mESC in the presence of 8 μg/ml Polybrene (Millipore, TR-1003-G) and centrifuged at 3000 rpm at 37 °C for 90 min. mESC were transferred back to a 37 °C, 5% $CO_2$ incubator for another 4 h before replacing the media with fresh mESC media. Antibiotic selection was initiated 36 h post-transduction.

## Flow cytometry

For detection of surface markers, dissociated mESC were first labeled with the eBioscience™ Fixable Viability Dye eFluor™ 780 (Thermo-Fisher, 65-0865-14) and then stained for surface markers with indicated fluorochrome-conjugated antibodies diluted in FACS buffer (PBS 1×, 0.5% BSA, 0.01% NaN₃): SSEA-1 (BD Biosciences, 562705), CD90.2 (Biolegend, 140319) and CD326 (Biolegend, 118227). For detection of intracellular markers, mESC stained with surface markers were subsequently fixed in PBS 1X + PFA 4%, rinsed with FACS buffer (PBS 1×, 0.5% BSA), and permeabilized in PBS 1X with 0.1% Triton X-100 for 5–10 min. For 5mC and 5hmC staining, the genomic DNA was denatured by treating the fixed mESC with 3.5 N HCl for 20 min. Following the removal of HCl, the pH was neutralized with 100 mM Tris-HCl (pH 8.5) for 10 min. at room temperature. mESC were rinsed in FACS buffer, blocked in blocking buffer (PBS 1X, 2% BSA, 2% Normal Donkey Serum (Sigma-Aldrich, d9663), and 0.02% Tween-20 (Sigma, P9416)), prior to the addition of the primary antibody (Active Motif, 39769). mESC were washed thrice with an intracellular wash buffer (PBS 1×, 0.5% BSA, and 0.02% Tween-20), before adding the secondary antibody. Acquisition was performed using a BD LSR Fortessa (BD Biosciences) and the BD FACSDiva software at the LJI Flow Cytometry Core Facility. All flow data were analyzed with FlowJo software (FlowJo LLC, Ashland, OR, USA).

## Immunocytochemistry of 8-cell embryos

8-cell stage embryos (E2.5) were flushed out of the oviduct by inserting a 31-gauge needle with a blunt end through the infundibulum and inside the oviduct. Harvested embryos were briefly washed in M2 medium, then the zona pellucida was removed by treatment with Acidic Tyrodes solution Serum (Sigma-Aldrich, T1788). Subsequently, the embryos were transferred on Denhart's coated coverslips and dried for 30 min. Embryos were then fixed for 15 min in 4% paraformaldehyde in PBS (EMS, 15713) at room temperature and permeabilized with 0.2% Triton X-100 (Sigma-Aldrich, T8787) in PBS for 5-

10 min at RT. The fixed embryos were blocked 1 hour at RT in 2% BSA, 0.05% Tween-20 in PBS, and then incubated in the same blocking solution for 1.5 h at RT with anti-γH2AX (phosphorylated Ser 139)-Alexa Fluor 647 (Biolegend, 613408). After several washes and Hoechst staining, embryos were mounted on slides with a small drop of Vectashield antifade mounting medium (Vector Laboratories, H-1000-10) or ProLong™ Diamond antifade mountant (ThermoFisher, P36961). Images were acquired using an Olympus FluoView FV10i laser scanning confocal microscope or the Zeiss LSM 880 with Airyscan microscope. Data analysis was performed with ImageJ software or Zeiss ZEN softwares.

### Fluorescence live-cell imaging

To study chromosome segregation by live-cell imaging, mESC were seeded on mitotically inactivated MEFs into a 96-well high optical quality plastic plates (Greiner Bio-One) and imaged using a Confocal Quantitative Image Cytometer CQ1 benchtop high-content analysis system (Yokogawa) with a $40 \times 0.95$ NA U-PlanApo objective and $2560 \times 2{,}160$-pixel sCMOS camera at $2 \times 2$ binning. 20–30 fields/well with $6 \times 2\,\mu m$ z-sections per field in GFP channel were captured at 6-minute intervals for 48 h. Chromosome segregation was manually analyzed using ImageJ.

### Metaphase Spread and G-banding

mESC (*Ctrl* and *Tet1/2/3 Tfl*) were exposed to 4-OHT for 2 days, then grown for an additional 4.5 days in SRES + 2i media at 37 °C in 5% $CO_2$, then treated with KaryoMAX™ Colcemid™ solution (Thermo Fisher Scientific, 15210040) at a final concentration of 100 ng/mL for 1 h at 37 °C in 5% $CO_2$ just before harvest and sorting by FACS using a FACSAria cell sorter (BD Biosciences). Sorted mESC were swelled in pre-warmed buffered hypotonic solution (Hepes 5 mM + KCl 65 mM) for 12 min at 37 °C, centrifuged to remove the hypotonic buffer before fixing them in fixative solution (methanol:glacial acetic acid, 3:1). The fixative solution was changed twice before storing the mESC at −20 °C overnight. On the next day, cells were resuspended in a small volume of fixative solution following two rinses. 30 µL of cell suspension were dropped onto clean dry glass slides, that were immediately exposed, face up, into the stream of hot steam (90 °C) for 30 seconds, which caused the cells to blow up as the fixative solution evaporated. Metaphase spreads were aged at room temperature for -10–15 days before performing G-banding. Suitably aged slides were incubated in 2X SSC at 65 °C for 1.5 h. Slides were transferred into 0.85% (w/v) NaCl at room temperature for 5 min, then into 0.85% NaCl + 0.025% trypsin for 15–20 s. Tryptic activity was stopped by placing the slides back into 0.85% NaCl, followed by 2 rinses in phosphate buffer 20 mM (pH 6.8). Slides were stained with fresh KaryoMAX™ Giemsa stain solution (Thermo Fisher Scientific, 10092013) in 5 mM phosphate buffer (pH 6.8) for 10 min. Slides were quickly rinsed again in phosphate buffer (pH 6.8) and blow-dried. Slides were mounted with Permount (Fisher Chemical, SP15-100) and photos were taken with a 100X oil-immersion lens.

### Quantitative real-time PCR assay

Total RNA was extracted with Quick-RNA Kit (Zymo Research, R1054) according to the manufacturer's protocol and reverse transcribed using SuperScript III reverse transcriptase and oligo(dT) primers (Thermo Fisher Scientific, 18080051). Synthesized cDNA was quantified with FastStart Universal SYBR® Green PCR Master Mix (Roche, 4913914001) and Step one real time PCR system (Thermo Fisher Scientific). The expression levels of target genes were normalized to the amount of *Gapdh* expression using the delta-delta CT method. Primers used in the analyses were as follows. *Khdc2* forward: 5′-AAA TAG AGT GGA TGT GCC AAG C-3′; *Khdc2* reverse: 5′-GGC CGC CAT GTT CAA GAG AA-3′; *Gapdh* forward: 5′-GTG TTC CTA CCC CCA ATG TGT-3′; *Gapdh* forward: 5′-ATT GTC ATA CCA GGA AAT GAG CTT-3′. GraphPad

Prism Version 9 was used to plot graphs, mean with standard deviation or error was shown on each graph.

### Western blot

Whole-cell extracts (WCE) were prepared by incubating mESC with radioimmunoprecipitation assay (RIPA) buffer (Thermo Fisher, 89900) supplemented with Benzonase (Sigma-Aldrich, E1014-25KU) and Halt™ Protease/Phosphatase Inhibitor Cocktail (ThermoFisher, 78441) on ice for 1 h. Proteins from WCEs were resolved using NuPAGE 4–12% bis-tris gel (Thermo Fisher Scientific, NP0321BOX) and transferred onto polyvinylidene difluoride (PVDF) membranes using Wet/Tank Blotting Systems (Bio-Rad). PVDF membranes were blocked with 5% nonfat milk in PBST (PBS 1×, 0.05% Tween-20) and incubated with the indicated primary antibodies, followed by secondary antibodies conjugated with horseradish peroxidase (HRP) (Cell Signaling, 7076 V), all of which were diluted 5% nonfat milk in PBST. Signal was detected with Femto Supersignal ECL substrate (ThermoFisher, 34096) and X-ray films (ThermoFisher, 34090).

### Whole genome sequencing (WGS) library preparation and mapping

Genomic DNA was isolated from bulk *Ctrl* and *Tet Tfl* mESC cells 6.5 days after 4-OHT treatment using the PureLink™ genomic DNA mini kit. Genomic DNA was fragmented to an average size of 400 bp using the Adaptive Focused Acoustics Covaris S2 instrument. Libraries were prepared using the TruSeq DNA PCR-Free Sample Preparation kit (Illumina, 20015963) according to the manufacturer's guidelines. The libraries were sequenced in an Illumina HiSeq 2500 instrument using single-end reads. Adapters and low-quality bases were trimmed before mapping and reads with length ≥ 70 bp were mapped to the *Mus musculus* genome (mm10 downloaded from UCSC website) using BWA-mem[89] with default options. Optical duplicate reads were removed Picard MarkDuplicates tool.

### Copy number analysis in whole-genome sequencing (WGS) and whole-genome bisulfite (WGBS) data

Low-coverage WGS libraries were sequenced on the Illumina Hiseq 2500 with the purpose to identify copy number alterations such as aneuploidies. Adapters and low-quality bases were trimmed before mapping and reads with length ≥ 70 bp were mapped to the *Mus musculus* genome (mm10, downloaded from UCSC website) using BWA-mem[89] with default options. High-coverage WGBS data was also mapped to *mm10* reference genome, using BSmap 2.74 v allowing for maximum four mismatches (-v 4) and removing low-quality 3′ reads (-q 10). Optical duplicate reads were removed Picard MarkDuplicates tool. HMMcopy[90] and CNVkit[91] were employed to detect copy number alterations, in WGS and WGBS, respectively, correcting for CG content and mappability bias, employing 500 kb bins (or as indicated).

### RNA extraction and RNA-seq libraries preparation

Total RNA was extracted from sorted Live/Dead dye$^-$/EGFP$^+$/CD90.2$^-$ mESCs with Quick-DNA/RNA Miniprep Kit (Zymo Research, D7001) according to the manufacturer's protocol. All RNAs were DNAse I treated in solution using the RNA Clean & Concentrator kit (Zymo Research, R1013) in order to eliminate any trace of genomic DNA from Total RNA. The yield and quality (RIN > 9.5 for all samples) of the purified RNAs were assessed using Qubit™ RNA HS Assay Kit (Thermo Fisher Scientific, Q32855) and TapeStation High Sensitivity RNA ScreenTape Analysis (Agilent, 5067-5579). RNA-Seq libraries were prepared using the Smart-Seq2 protocol[92], as have been published before[93]. All libraries were assessed using Qubit™ RNA HS Assay Kit (ThermoFisher, Q32855) and TapeStation High Sensitivity RNA ScreenTape Analysis (Agilent, 5067-5579). 20 million paired-end reads (50 × 50 bases) were generated from each RNA-Seq library using a NovaSeq sequencing platform (Illumina).

## Bioinformatic analyses

For RNA-seq, reads were aligned using STAR[94] with the parameters --outFilterMultimapNmax 1, --outSAMtype BAM SortedByCoordinate –sjdbOverhang 100. HT-Seq[95] was used to quantify the gene expression levels using the options htseq-count -s yes, -r pos, -a 10. Normalization and differential expression analyses were performed using DESEq2[96], with the parameters fitType parametric, alpha 0.05 and using the Benjamini & Hochberg method. For visualization of the data, we generated tracks using Deeptools[97], with the option bamCoverage. All related plots were made using R-Studio[98] and Integrative Genome Viewer (IGV)[99].

For ChIP-Seq datasets, we used Bowtie[100] for the alignments and Deeptools with the option bamCoverage for genome track generations. For analyses of DNA methylation status on wild-type and *Tet TKO* mESC we used Whole Genome Bisulfate sequencing (WGBS) samples we used Bismark[101] and bedGraphToBigWig for tracks generation.

## Reporting summary

Further information on research design is available in the Nature Research Reporting Summary linked to this article.

## Data availability

The authors confirm that the data supporting the conclusions of this study are included in this published article and its supplementary files (including source data files). The sequencing datasets generated in this study have been deposited in the NCBI Gene Expression Omnibus and are accessible through the GEO Series accession number GSE191045 for RNA-seq and GSE214402 for low coverage WGS from bulk samples. Other published datasets used on this study include: GSE77781[62], GSE57413[63], GSE57700[64] and GSE115972[65] for ChIP-Seq data; GSE56986[51], GSE116482[61], GSE126958[67], GSE72856[71], GSE158460[72], GSE48519[102], GSE63281[103] and GSE116420 for WGBS; GSE56986[51], GSE95720[59], GSE122814[60], GSE116482[61] and GSE48519[102] for RNA-Seq. Source data files are provided with this paper. Source data are provided with this paper.

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

## Acknowledgements

We thank all members of the Rao laboratory for suggestions and discussions. We thank C. Kim, C. Dillingham, D. Hinz, L. Boggeman, and M. Haynes at the La Jolla Institute Flow Cytometry facility for help with cell sorting experiments and J. Day of the La Jolla Institute Sequencing facility for help with next-generation sequencing, Z. Mikulski of the La Jolla Institute Microscopy facility for help with microscopy and D. Jenkins of the Small Molecule Discovery Program at the Ludwig Institute for Cancer Research for help with live-cell imaging. We also thank Dr. Margaret Goodell (Baylor) and Dr. Yun Nancy Huang (Texas A&M) for providing the *Dnmt TKO* mESC. We thank Dr. Jinsuk Kang for help with preparing slides for IFC staining on early embryos. This work was supported by NIH grants R35 CA210043 to A.R., R01 GM074215 to A.D. BD FACSAria II is supported by NIH (NIH S10OD016262, NIH S10RR027366), and our research used resources of the Advanced Light Source, which is a DOE Office of Science User Facility under contract no. DE-AC02-05CH11231. The NovaSeq 6000 and the HiSeq 2500 were acquired through the Shared Instrumentation Grant (SIG) Program (S10); NovaSeq 6000 S10OD025052 and HiSeq 2500 S10OD016262. R.G. was supported by T32 (AI125179) and F32 postdoctoral fellowships from NIH; H.S. was supported by the Pew Latin-American Fellows Program from The Pew Charitable Trusts. I.F.L.-M. was supported by a University of California Institute for Mexico and the United States–Consejo Nacional de Ciencia y Tecnología (UC MEXUS-CONACYT) Fellowship.

## Author contributions

R.O.G. generated the *Tet iTKO* mice system and derived the mESCs and acquired the data. H.S., J.C.A., and I.F.L.-M. performed the bioinformatics analyses. R.O.G., A.R., A.D., and H.S. interpreted the data. A.D. provided the instruments and protocols for live-cell imaging. E.J., S.P., and R.B.N. helped in the imaging data analysis. R.O.G., H.S., I.F.L.-M., and A.R. wrote the manuscript. All authors were involved in reviewing and editing the manuscript.

## Competing interests

The authors declare no competing interests.
