## [Peer Review File · Nature Communications]

Acute deletion of TET enzymes results in aneuploidy in mouse embryonic stem cells through decreased expression of Khdc3.Editorial Note: Parts of this Peer Review File have been redacted as indicated to remove third-party material where no permission to publish could be obtained.

REVIEWER COMMENTS

Reviewer #1 (Remarks to the Author):

Georges et al have developed an elegant system for rapid recombination and deletion of TET1/2/3 as well as an H2B-EGFP reporter in embryonic stem cells and combined this with short-term 4-OHT treatment and FACS sorting to obtain pure populations of TET TKO cells. The subsequent analysis of metaphase spreads of the polyclonal populations suggest that acute TET triple knockout leads to increased aneuploidy at least partly through the downregulation of Khdc3 expression (a gene known to play a role in the SCMC in embryonic stem cells). To link this phenotype to early development, the authors furthermore examine embryogenesis using Stra8 and Zp3-driven Cre transgenes and shows an increase in abnormal blastomeres in TET triple knockout embryos (although this analysis is complicated by confounding factors of the early embryonic lethality observed previously). As such, the authors employ an elegant model system to analyse the early effects of TET triple deletion in embryogenesis. The study includes essential controls and represents an important advancement in our understanding of the role of TET enzymes and DNA methylation in early development and genome stability.

However, before recommending the publication of this manuscript I believe that there are important aspects that need to be addressed as outlined below:

Major concerns

The main observation of the manuscript is the increase in aneuploid cells observed as early as 6.5 days after induction of recombination. While this could be a result of loss of TET proteins, as suggested by the authors, it may also be result of aberrant chromosomal translocations that occur between LoxP sites located at different chromosomes (Ctrl cells contain two LoxP sites, while TET TKO cells contains 14) which in turn would lead to mis-segregation and aneuploidy in daughter cells after subsequent cell divisions. To formally exclude this, it would be necessary to obtain evidence using one of the two options listed below: i) Generate control cells with similar number of loxP sites that delete unrelated control regions or genes with no known role in chromosome segregation or ii) Use other means to downregulate expression of the TET enzymes (e.g. shRNA or CRISPRi) to validate that aneuploidy is induced by downregulation of TET enzymes in the absence of extensive Cre-induced recombination.

In Figure 3, the authors describe an experiment in which control and Tet TKO cells have been subcloned and passaged for 1 month prior to analysis of metaphase spreads. If the increased aneuploidy is a result of loss of TET proteins and downregulation of Khdc3, it is reasonable to expect that long-term passaging of the cells would result in an increased likelihood of mis-segregation of chromosomes at every cell cycle and a progressive departure from euploidy over time. However, the results presented in Figure 3B suggests that the fraction of euploid cells (~50%) are maintained in TET TKO cells even in the absence of the TET enzymes and Khdc3 expression. Thus, it is possible that the initial increase in aneuploidy is a result of aberrant LoxP site recombination and not the loss of TET enzymes (as described above). To lend more support to this point, it would be helpful to do a more extensive clonal analysis of aneuploidy (e.g. pick two clones of each genotype and subclone these into three sublines to identify the basal level of aneuploidy as well as the departure from baseline upon extended passaging).

In Figure 5, the authors show that overexpression of Khdc3 is sufficient to rescue the effect of TET TKO in inducing aneuploidy. While it is a striking observation that overexpression of a single gene can rescue the effect (several other genes involved in chromosome segregation are found to be misregulated), it would lend further support to the hypothesis to examine the effects of Khdc3 knockdown or knockout on chromosome stability in this particular ES cell line. Is knockdown of Khdc3 (in the absence of 4-OHT treatment and Cre recombination) sufficient to increase aneuploidy in the ES cell lines in question?

Minor concerns

From the data presented it is not possible to assess if the sorting of EGFP+ cells after short-term 4-OHT treatment leads to a selection of a specific subset of cells within the ES cell population which is particularly responsive to the treatment. It would be useful that the authors provide FACS plots to show the distribution of EGFP+ cells compared to the surface markers analysed (e.g. SSEA, CD90, and CD326).

The results suggest that loss of TET enzymes leads to decreased expression of Khdc3, presumably through the combined action of TET1 and TET2 (TET3 is low or absent in these cells). However, it would be useful to examine if Khdc3 expression is affected in ES cells lacking either TET1 or TET2 alone to relate these results to existing studies of embryonic stem cells as well as to hematological malignancies where often only one member of the TET enzyme family is mutated or downregulated. The authors could generate and examine TET single knockout cells themselves or use existing datasets to explore if Khdc3 might be deregulated in this context. In light of this analysis, it would be interesting to relate the findings to the apparent unaffected fertility and perinatal viability of TET1 and TET2 single knockout mice lines?

Reviewer #2 (Remarks to the Author):

The authors here describe an interesting aneuploidy phenotype upon TET1/2/3 depletion in mESCs. More studies are needed to connect this phenotype to the functions of TET1/2/3.

Major comments:

1. It is unclear that acute deletion of Tet1/2/3 seems to have a different phenotype from regular deletion of Tet1/2/3 in mESCs (two references [Tet inactivation disrupts YY1 binding and long-range chromatin interactions during embryonic heart development] and [Deletion of Tet proteins results in quantitative disparities during ESC differentiation partially attributable to alterations in gene expression]). Is this due to the 'acute' deletion way or others? Also, acute deletion of the genes does not necessarily result in acute depletion of proteins – proteins can still retain for ~5 days after genes deletion (which is almost the same to the other non-acute methods). However, it will be very interesting to analyze the differences between different TET1/2/3 depletion methods and get insight on the new potential compensation or competition between TET1/2/3.

2. Given the differences resulted from different TET1/2/3 depletion methods, it is necessary to examine the whole genome DNA 5mC and/or 5hmC in the current Tet1/2/3 TKO cell lines. Combining the 5mC/5hmC profiles and RNA-seq profiles, we can get a better understanding of the aneuploidy phenotype and further confirmation of Khdc mechanism.

3. Studies of the early embryos are not quite clear either. It is known that maternal Tet3 deletion can cause failure of paternal pronuclear DNA demethylation. Is this also observed in the TKO embryos here? It is better to provide a whole picture of early cleavage stages from 1C to morula, systematically count the micronuclei number at every stage. Also, please include the bright field images of the embryos. It is confusing that TKO embryos had more micronuclei and nuclei under mitosis, but the number of blastomeres had such a big deviation to both increase and decrease. Does this mean some of the aneuploid cells can separate well but others cannot?

4. It is not clear if the authors are trying to connect the results of Figure 5 and supp Figure 4 to imply a potential TET mediated mechanism in a long-term aneuploidy in malignancy. Currently the Supp figure 4 looks disconnected. Some additional work is needed to further support this hypothetical link. For example, a whole genome study of patient samples (or analyze the sequencing files available), or a differentiation study of the long-term cultured TKO cells to indicate a higher capacity to transform,

or identification of some potential hot spots from the patients' genome in the long-culture TKO cells showing aneuploidy, etc.

Minor comments:

1. "Deletion" and "depletion" are usually used specifically to regard either DNA level knockout or protein level clearance. The two words were not used properly for multiple times in the manuscript, including title, results and figure legends. Please try to correct them into proper ways: for example, delete Tet 1/2/3 genes, Tet1/2/3 deletion, depletion of TET1/2/3 enzymes.
2. In page 3, it is unreasonable to say 'aneuploidy was not observed in Tet Tfl cells not treated with tamoxifen' given the 20% background aneuploidy in the figure.
3. In figure 4, the volcano plot and the MA-plot are talking about the same thing. Usually people choose one plot type to present.
4. In supp figure 2, why the flag-KHDC3 overexpression has so many bands?
5. As the aneuploidy increases in long period culture of TKO cells, will KHDC3 overexpression still be able to stop further accumulation of aneuploidy or partially rescue the aneuploidy?
6. There is a mistake in using HTseq to count RNA-seq reads. Smart-seq2 protocol can not give the stranded information of the libraries. So "htseq-count -s no" should be used instead of "htseq-count -s yes".
7. There is no description in the methods about how the authors evaluated aneuploidy using their WGS data. The labeling of Supp Figure 1 is a little confusing -- why the y-axis is tumor copy number? Is this the copy ratio of a specific set of tumor genes?
8. Just for curiosity—if doing WGS of the long-term-cultured TKO cells, is that possible to observe aneuploidy?

Reviewer #3 (Remarks to the Author):

The effect of Tet deficiency on ES cells has not been fully unexplored. It is nice that the authors have taken new effort to characterize the immediate consequence of Tet deletion. The observation of aneuploidy in Tet TKO ES cells is interesting. Overall, the data presented agree well with the conclusion that downregulation of Khdc3 might cause the aneuploidy phenotype. Of note, the inclusion of control mESCs in Fig 5B is laudable. However, the manuscript would be improved to be suitable for publication if the issues raised below could be addressed.

- 1) Are related chromosomal abnormality observed in Dnmt TKO ES cells? Which of the DNA methyltransferases is responsible for the gain of methylation at Khdc3 in Tet triple knockout ESCs? Upon the deletion of the responsible Dnmt, the effect of aneuploidy etc should be reversed. This rescue assay is highly recommended to complement the data shown in Fig 5. Dnmt TKO appears to have less severe effect on mESCs and this makes a rescue experiment by Dnmt deletion feasible.
- 2) Fig. 5, the protein expression levels of lentiviral Khdc should be carefully controlled to make sure that they are equivalent of the levels of endogenous proteins in wildtype ESCs. Over-expression (OE as indicated in the figure) could lead to an over-rescue thus overlooking other potential mechanisms contributing to the aneuploidy phenotype.
- 3) Transcriptional profiling of Tet TKO preimplantation embryos would be important to compare the in

vitro observations in ESCs with in vivo effects. One question is whether Khdc3 would also be downregulated, concomitantly with increased methylation of CpG dinucleotides in the vicinity of the Khdc3 gene in the TKO embryos.

Other points:

1. Comparison of Tet TKO mESCs that have been maintained in culture, with mESCs of acute Tet deletion in the same experiment would be interesting. How is the change at Khdc3 in long-term TKO mESCs?
2. It should be explained why 5' exons of Tet1 and Tet2 were chosen for deletion rather than 3' exons encoding the catalytic domains. Since the 3' exons are expressed normally (Fig 1b), it is necessary to ensure that no truncated proteins of residual enzymatic activity can be generated in iTKO cells. Would the deletion of Tet1 and Tet2 exons cause frame-shift mutations?
- 3 . Which of the Tet enzymes (Tet1 versus Tet2) is more important for maintaining a hypomethylated state for Khdc3?
4. Tet3 is known to be not expressed in mouse ES cells. Could the expression observed in Fig 1B suggest that the cultured iTKO ES cells have undergone a certain degree of differentiation?
5. Fig 2, would aneuploidy occur also in Dnmt TKO mESCs? This data would be desirable to be included in Fig 2D, if possible.

Reviewer #1 (Remarks to the Author):

Georges et al have developed an elegant system for rapid recombination and deletion of TET1/2/3 as well as an H2B-EGFP reporter in embryonic stem cells and combined this with short-term 4-OHT treatment and FACS sorting to obtain pure populations of TET TKO cells. The subsequent analysis of metaphase spreads of the polyclonal populations suggest that acute TET triple knockout leads to increased aneuploidy at least partly through the downregulation of *Khdc3* expression (a gene known to play a role in the SCMC in embryonic stem cells). To link this phenotype to early development, the authors furthermore examine embryogenesis using *Stra8* and *Zp3*-driven *Cre* transgenes and shows an increase in abnormal blastomeres in TET triple knockout embryos (although this analysis is complicated by confounding factors of the early embryonic lethality observed previously). As such, the authors employ an elegant model system to analyse the early effects of TET triple deletion in embryogenesis. The study includes essential controls and represents an important advancement in our understanding of the role of TET enzymes and DNA methylation in early development and genome stability.

We thank the reviewer for these positive comments.

However, before recommending the publication of this manuscript I believe that there are important aspects that need to be addressed as outlined below. Major concerns

The main observation of the manuscript is the increase in aneuploid cells observed as early as 6.5 days after induction of recombination. While this could be a result of loss of TET proteins, as suggested by the authors, it may also be result of aberrant chromosomal translocations that occur between *LoxP* sites located at different chromosomes (*Ctrl* cells contain two *LoxP* sites, while TET TKO cells contains 14) which in turn would lead to mis-segregation and aneuploidy in daughter cells after subsequent cell divisions. To formally exclude this, it would be necessary to obtain evidence using one of the two options listed below: i) Generate control cells with similar number of *loxP* sites that delete unrelated control regions or genes with no known role in chromosome segregation or ii) Use other means to downregulate expression of the TET enzymes (e.g. shRNA or CRISPRi) to validate that aneuploidy is induced by downregulation of TET enzymes in the absence of extensive *Cre*-induced recombination.

We agree with the reviewer that it would be important to repeat our experiments using strategies that do not involve *Cre*-mediated deletion. We chose not to do this, however, because such experiments have already been performed by other groups in the same cellular system, mouse embryonic stem cells. Yang *et al.*, 2016 [1] used stable retroviral shRNA-mediated depletion of *Tet1* and *Tet2* in mESC as well as previously generated *Tet1/2* DKO mESC, and observed chromosomal fusions and telomere shortening, while Kafer *et al.*, 2016 [2] used CRISPR/Cas9 technology to disrupt the expression of all 3 *Tet* genes in mESC and observed increased frequencies of mitotic abnormalities. Because these gene silencing (RNAi) and gene editing (CRISPR) strategies required prolonged periods of culture, we focused on the acute deletion system shown in Figure 1, with the goal of asking whether the emergence of genomic instability was an early and direct event caused by TET loss-of-function and discovering the mechanism involved. Moreover, we confirmed the genomic instability observed in the papers cited above (which used RNAi and CRISPR/Cas9 deletion) by showing that the genomic instability observed in acutely deleted *Tet iTKO* mESCs was maintained after prolonged proliferation (>70 cell divisions). To do this, we derived clones of “long-term” *Tet iTKO* mESC by sorting single cells from the parental acutely deleted *Tet iTKO* mESC populations followed by expansion in culture. Since genomic instability is observed in both “acute” and “long-term” *Tet iTKO* mESCs generated in our study, we feel confident that the chromosome mis-segregation phenotype is a direct and immediate consequence of TET loss-of-function.

We also agree with the reviewer that recombination among multiple *loxP* sites, within mESCs harboring floxed alleles of all 3 *Tet* genes, could in principle be associated with increased aneuploidy secondary to aberrant chromosomal translocations due to *Cre*-mediated double strand DNA breaks. However, this mechanism seems unlikely to be significant for several reasons that we have now included in the revised Discussion.

1. Re-expression of *Khdc3* in *Tet iTKO* mESC – in the same 6.5-day time period as 4-OHT treatment, *Cre* activation and *Tet* gene deletion – prevented the observed increase in aneuploidy, suggesting that aneuploidies triggered simply by *Cre* expression do not constitute a significant proportion of total aneuploidies in *Tet iTKO* mESC.
2. To minimize the effects of lingering *Cre*-ERT2 activity in the nucleus, we deliberately removed 4-OHT on day 2.5, 4.5 days before performing karyotype analysis on day 6.5-7.0. This strategy returns unliganded *Cre*-ERT2 to the cytoplasm, thus terminating its recombinase activity in the nucleus and preventing further recombination among potential cryptic *loxP* sites in the genome, the single *loxP* sites remaining within *Tet* genes, or both.
3. The frequency of chromosomal translocations occurring between *loxP* sites located in *trans* (on different chromosomes) is very low. The 3 *Tet* genes are located on 3 different chromosomes – chromosome arm 10q for

Tet1, chromosome arm 3q for *Tet2*, and chromosome arm 6q for *Tet3*. *Cre-loxP*-mediated recombination efficiency decreases with increasing genetic distance separating 2 *loxP* sites in *cis*, with an estimated efficiency of 11% for a 2 centiMorgan (cM) (4 Mb) separation between *loxP* sites and a 5 orders of magnitude lower efficiency (0.0001%) for a 60 cM separation (Zheng B. *et al* MCB 2000 [3]). The efficiency of *Cre-loxP*-mediated chromosomal translocations, occurring between *loxP* sites located on different chromosomes, is estimated to fall between these values (0.04-0.08%; ~1 *Cre*-mediated translocation in 1200-2400 mESC constitutively expressing the *Cre* recombinase) (Zheng B. *et al* MCB 2000 [3]; Van Deursen J. *et al* PNAS [4]). Thus, relative to *Cre-loxP*-mediated recombination between the neighboring *loxP* sites located in *cis* within the *Tet* genes, *Cre-loxP*-mediated recombination in *trans* is expected to be a rare event with only a minor effect, if any, on the increase in aneuploid frequency of *Tet iTKO* mESCs (see revised Discussion in the new version of our Manuscript).

- Even if recombination occurred between single *loxP* sites generated after deletion of floxed *Tet* exons located on different chromosomes, chromosomal translocation induced by *Cre-loxP* recombination is usually accompanied by the generation of dicentric chromosomes, which we never observed in our live-imaging experiments and observed very rarely in our metaphase spreads. The absence of dicentric chromosomes in *Tet iTKO* mESCs suggests that the aneuploidy observed in the *Tet iTKO* mESCs is not a consequence of *Cre-loxP* recombination in *trans*.

In Figure 3, the authors describe an experiment in which control and *Tet* TKO cells have been subcloned and

and passaged for 1 month prior to analysis of metaphase spreads. If the increased aneuploidy is a result of loss of TET proteins and downregulation of *Khdc3*, it is reasonable to expect that long-term passaging of the cells would result in an increased likelihood of mis-segregation of chromosomes at every cell cycle and a progressive departure from euploidy over time. However, the results presented in Figure 3B suggests that the fraction of euploid cells (~50%) are maintained in TET TKO cells even in the absence of the TET enzymes and *Khdc3* expression. Thus, it is possible that the initial increase in aneuploidy is a result of aberrant *LoxP* site recombination and not the loss of TET enzymes (as described above). To lend more support to this point, it would be helpful to do a more extensive clonal analysis of aneuploidy (e.g. pick two clones of each genotype and subclone these into three sublines to identify the basal level of aneuploidy as well as the departure from baseline upon extended passaging).

As described in our response above, “long-term” *Tet iTKO* mESCs are in fact monoclonal populations sorted and subcloned from parental “acutely deleted” *Tet iTKO* mESC. The sorting and cloning of these cells necessarily required maintaining them in culture for extended times. We have emphasized this point in the revised Discussion.

As the reviewer notes, acutely-deleted and long-term clonal *Tet iTKO* mESCs

displayed equivalent frequencies of aneuploidy. The reviewer's interpretation seems to be that *Tet iTKO* mESC lines would necessarily show increased aneuploidy after long-term culture, but this interpretation does not take into account the well-known selective pressures that oppose the development of aneuploidy in most normal cell types; in fact, the large majority of chromosome gains and losses are known to impair cell fitness (reviewed in Santaguida S. and Amon A. *Nat Rev Mol Cell Biol* 2015 [5] and Chiarle R. *Genes Dev.* 2021 [6]). We assume, therefore, that the increased aneuploidy that we observe at a single-cell level (by metaphase spreads) in both acute and long-term *Tet iTKO* mESCs cultured in 2i media involve chromosomes whose gains or losses do not seriously hamper mESC proliferation and/or survival, explaining the relative stability of the aneuploid phenotype following *Tet* gene deletion. In this context, we note that random chromosome gains and losses are observed early in cancer development, because selective pressures have not yet come into play; at later stages, specific chromosomal gains and losses are recurrently selected for, presumably because they confer increased fitness to and promote expansion of those cancer clones (Shoshani O. *et al Genes Dev.* 2021 [7]; Trakala M. *et al Genes Dev.* 2021 [8]); reviewed in Chiarle R. *Genes Dev.* 2021 [6]). We have emphasized these points in the revised Discussion.

In Figure 5, the authors show that overexpression of *Khdc3* is sufficient to rescue the effect of TET TKO in inducing aneuploidy. (i) While it is a striking observation that overexpression of a single gene can rescue the effect (several other genes involved in chromosome segregation are found to be misregulated), it would lend further support to the hypothesis to examine the effects of *Khdc3* knockdown or knockout on chromosome stability in this particular ES cell line. (ii) Is knockdown of *Khdc3* (in the absence of 4-OHT treatment and Cre recombination) sufficient to increase aneuploidy in the ES cell lines in question?

[redacted]

(i) Among the subset of mis-regulated genes involved in chromosome segregation in *Tet iTKO* mESCs, *Khdc3* is the most drastically altered (see **Fig. 3**), which might explain why re-expression of this single gene product allows rescue of increased aneuploidy. In contrast, down-regulation of *Khdc2* is not significant in “long-term” *Tet iTKO* mESCs (please see **NEW Suppl. Fig. 4D**), indicating that the mild decrease of *Khdc2* expression in acutely-deleted *Tet iTKO* mESCs is not essential for the emergence of aneuploidies following TET loss-of-function. (ii) We agree that validation of the effects of *Khdc3* knockdown or knockout on chromosome stability in our own ESC system would be beneficial to our study and have added a figure showing that CRISPR/Cas9-mediated disruption of either the *Khdc2* or *Khdc3* genes results in increases aneuploidy in mESCs, even if expression of the TET enzymes is maintained (**NEW Suppl. Fig. 9**). In that

regard, our results confirm a previous report (Zhao Bo *et al Cell Stem Cell* 2015 [9]) that also shows clearly that deletion of *Khdc3* in mESC results in aneuploidy.

Minor concerns

From the data presented it is not possible to assess if the sorting of EGFP+ cells after short-term 4-OHT treatment leads to a selection of a specific subset of cells within the ES cell population which is particularly responsive to the treatment. It would be useful that the authors provide FACS plots to show the distribution of EGFP+ cells compared to the surface markers analyzed (e.g. SSEA, CD90, and CD326/EPCAM).

Our mESCs were grown on a feeder layer of mitotically inactive mouse embryonic fibroblasts (MEFs), thus we stained cells with cell surface markers specific for mESCs (SSEA1, SSEA-1, CD326 (Epcam)) or MEF (CD90, Thy1.2+) as well as H2B-GFP. We have now included detailed flow cytometry plots to show the staining patterns (**NEW Suppl Figure 1**). EGFP+ cells represent >95% of the cells following 4-OHT treatment (from Day 1.5 to Day 6.5).

The results suggest that loss of TET enzymes leads to decreased expression of *Khdc3*, presumably through the combined action of TET1 and TET2 (TET3 is low or absent in these cells). However, it would be useful to examine if *Khdc3* expression is affected in ES cells lacking either TET1 or TET2 alone to relate these results to existing studies of embryonic stem cells as well as to hematological malignancies where often only one member of the TET enzyme family is mutated or downregulated. i) The authors could generate and examine TET single knockout cells themselves or use existing datasets to explore if *Khdc3* might be deregulated in this context.

We examined previously published RNA-Seq and WGBS datasets from single *Tet1*- and *Tet2*-deficient mESC [9] and found that *Tet2*-deficient mESC display a mild reduction of *Khdc3* expression, as well as a slight increase in DNA methylation of the *Khdc3* promoter suggesting that Tet2 has a more critical role in regulating *Khdc3* expression. The data have been added to the **NEW Supplementary Figure 8**. This is consistent with higher occupancy of the *Khdc3* promoter by Tet2, compared to Tet1 (*ChIP-seq* data, Fig 4A).

ii) *In light of this analysis, it would be interesting to relate the findings to the apparent unaffected fertility and perinatal viability of TET1 and TET2 single knockout mice lines?*

We thank the reviewer for this suggestion. Indeed, partial compensation among *Tet* genes explains the more severe developmental abnormalities and embryonic lethality of compound *Tet* mutant mice [11-15]. We have added a few sentences on this subject to the revised Introduction.

References for reviewer 1.

1. Yang J. *et al.*, Cell Reports 2015. [PMID: 27184841]
2. Kafer G. *et al.*, Cell Reports 2016 [PMID: 26854228]
3. Zheng B. *et al.* MCB 2000 [PMID: 10611243]
4. Van Deursen J. *et al* PNAS 1995 [PMID: 7638200]
5. Santaguida S. and Amon A. Nat Rev Mol Cell Biol 2015 [PMID: 26204159]
6. Chiarle R. Genes Dev. 2021 [PMID: 34341000]
7. Shoshani O. *et al* Genes Dev. 2021 [PMID: 34266887]; reviewed in [6]
8. Trakala M. *et al* Genes Dev. 2021 [PMID: 34266888]; reviewed in [6]
9. Zhao Bo *et al* Cell Stem Cell 2015 [PMID: 25936915]
10. Hon G. *et al* Mol Cell 2014 [PMID: 25263596]
11. Dawlaty MM. *et al* Cell Stem Cell 2011 [PMID: 21816367]
12. Dawlaty MM. *et al* Dev Cell 2013 [PMID: 23352810]
13. Kang J. *et al* PNAS 2015 [PMID: 26199412]
14. Li X. *et al* PNAS 2016 [PMID: 27930333]
15. Dai HQ *et al* Nature 2016 [PMID: 27760115]

Reviewer #2 (Remarks to the Author):

The authors here describe an interesting aneuploidy phenotype upon *TET1/2/3* depletion in mESCs. More studies are needed to connect this phenotype to the functions of *TET1/2/3*.

Major comments:

1. i) It is unclear that acute deletion of *Tet1/2/3* seems to have a different phenotype from regular deletion of *Tet1/2/3* in mESCs (two references [*Tet inactivation disrupts YY1 binding and long-range chromatin interactions during embryonic heart development*] and [*Deletion of Tet proteins results in quantitative disparities during ESC differentiation partially attributable to alterations in gene expression*]). Is this due to the 'acute' deletion way or others? ii) Also, acute deletion of the genes does not necessarily result in acute depletion of proteins – proteins can still retain for ~5 days after genes deletion (which is almost the same to the other non-acute methods). However, it will be very interesting to analyze the differences between different *TET1/2/3* depletion methods and get insight on the new potential compensation or competition between *TET1/2/3*.

i) The reviewer is correct: we demonstrate in this study that acute and long-term triple *TET* deletion result in similar levels of aneuploidy. Previous studies employed CRISPR-mediated disruption of *Tet* genes (e.g. Kafer GR *et al.*, 2019, PMID 26854228; Lu F. *et al* 2014, PMID 25223896; Wang H. *et al.*, 2013 [PMID 23643243], Reimer M *et al.*, 2019, PMID 31286885) or retroviral shRNA-mediated depletion of *Tet* mRNA and protein (e.g. Yang *et al.*, 2016, PMID 27184841). Because these methods require extended culture and multiple mESC passages to perform the necessary antibiotic selection steps and confirm gene disruption, they could potentially induce compensatory changes that overshadow the immediate consequences of *TET* loss-of-function. Hence, we chose to focus on acute *Tet* gene deletion, to ask whether genome instability was an early direct effect of *TET* loss-of-function. We have emphasized this point in the revised Discussion.

Of the two references cited by the reviewer, the first [PMID: 31541101] is focused on cardiac development in *Tet2*^{-/-}, *Tet3* *fl/fl*, *Nkx2.5-Cre* mice, although it includes one set of experiments performed in *TET TKO* mESC generated using CRISPR/Cas9 technology. The time period for which the mESCs were cultured after triple *TET* deletion was not noted, however. The second reference [PMID: 31286885] is specifically on *Tet* gene deletion in mESC, and therefore is more pertinent to our study. We thank the reviewer for bringing this study to our attention and have now referenced it in the text.

iii) We agree completely that gene disruption does not necessarily result in acute depletion of the encoded protein, and indeed took pains to show in **Figure 1** of the original manuscript that Tet1 and Tet2 are fully depleted at

the protein level 6 days after 4-OHT treatment. We now extend these findings in the **NEW Suppl. Fig. 2**, by showing that that Tet1 and Tet2 are already undetectable at the protein level by Day 1.5 after withdrawal of 4-OHT (i.e. 4 days after 4-OHT addition).

2. Given the differences resulted from different *TET1/2/3* depletion methods, it is necessary to examine the whole genome DNA 5mC and/or 5hmC in the current *Tet1/2/3* TKO cell lines. Combining the 5mC/5hmC profiles and RNA-seq profiles, we can get a better understanding of the aneuploidy phenotype and further confirmation of *Khdc* mechanism.

We have not examined 5hmC distribution in *Tet iTKO* mESC, since we showed in the original version of this manuscript that 5hmC levels fall to baseline levels within 6 days of 4-OHT addition (**Figures 1D, E**). As for alterations in 5mC distribution, the genome-wide methylome of triple *Tet*-deficient mESCs has been analyzed many times before (e.g. Lu F *et al.*, 2014, PMID: 25223896; Wang Q *et al.*, 2020, PMID: 32690947), and WGBS data from these and other studies consistently show that the *Khdc3* promoter is hypermethylated (**NEW Figure 4**). Moreover, our rescue experiments show unambiguously that the increased methylation of the *Khdc3* promoter and the associated downregulation of *Khdc3* gene expression account for the bulk of chromosome mis-segregation leading to increased aneuploidy in *Tet iTKO* mESCs. Thus, although differential methylation at the promoters of other target genes involved in chromosome segregation might contribute to the observed aneuploidy, yet another genome-wide analysis of 5hmC

distribution in mouse ES cells seems unnecessary and outside the focus of this study. We hope the reviewer and the editors agree.

3. Studies of the early embryos are not quite clear either. i) It is known that maternal *Tet3* deletion can cause failure of paternal pronuclear DNA demethylation. Is this also observed in the *TKO* embryos here? It is better to provide a whole picture of early cleavage stages from 1C to morula, systematically count the micronuclei number at every stage.

The scientific literature already contains many outstanding studies of the effects of single and combined *Tet* gene deletions in early mouse embryogenesis [Dai HQ. *et al* (2016) [PMID: 27760115]; Kang J. *et al* (2015) [PMID: 26199412]]. In light of these studies, we analyzed 8-cell embryos in the original Fig. 2F-G, with the single straightforward idea of emphasizing that the chromosome mis-segregation we observe in acutely deleted *Tet iTKO* mESC in culture is also apparent in *Tet TKO* mice *in vivo*. We hope the reviewer and editors agree that a more comprehensive analysis of embryonic development in *Tet TKO* mice is outside the scope of this manuscript.

Nevertheless, to address the reviewer's point, we examined two published RNA-seq datasets of *Tet*-deficient blastocysts [Dai HQ. *et al* (2016) [PMID: 27760115]; Kang J. *et al* (2015) [PMID: 26199412]] and found that *Khdc3* expression is indeed significantly perturbed when *Tet* enzymes are removed at an early stage of embryonic development. These data are now included in the **NEW Suppl. Fig. 7C-D**.

From **NEW Supplementary Figure 7C-D: The *Khdc3* gene is a conserved target of Tet proteins, that are required to sustain normal expression of *Khdc3* in mESC and in early development.** (C) RNAs from five *Tet1/3 DKO* blastocysts (3.5 days) were sequenced and compared to three wildtype (WT) blastocysts. 3 of 5 *Tet1/3 DKO* blastocysts failed to maintain normal levels of *Khdc3*. (D) Three blastocysts lacking all Tet enzymes (*Tet TKO*) were compared to the same number of control (WT) blastocysts. All *Tet TKO* blastocysts have reduced levels of *Khdc3* compared to their controls.

ii) Also, please include the bright field images of the embryos.

We have added supplementary panels that will include the bright field images of 8-cell stage embryos (*Ctrl* and *Tet TKO*) in **NEW Suppl. Fig. 5**.

iii) It is confusing that *TKO* embryos had more micronuclei and nuclei under mitosis, but the number of blastomeres had such a big deviation to both increase and decrease. Does this mean some of the aneuploid cells can separate well but others cannot?

The definition of aneuploidy – an abnormal number of chromosomes – encompasses both gains and losses of chromosomes. Based on the images of 8-cell stage embryos, the presence of an increased number of unevenly sized blastomeres might result from blastomere fragmentation; alternatively, the cell cycle may be slightly accelerated in some *Tet TKO* cells/ embryos, resulting in premature blastomere division and hence increased blastomere numbers. “8-cell stage” embryos with a

reduced number of blastomeres could reflect apoptotic death of a subset of blastomeres or fragmented blastomeres, as observed in some embryos. We have added this explanation to the Results section of the revised manuscript.

4. It is not clear if the authors are trying to connect the results of Figure 5 and supp Figure 4 to imply a potential TET mediated mechanism in a long-term aneuploidy in malignancy. Currently the Supp figure 4 looks disconnected. Some additional work is needed to further support this hypothetical link. For example, a whole genome study of patient samples (or analyze the sequencing files available), or a differentiation study of the long-term cultured *TKO* cells to indicate a higher capacity to transform, or identification of some potential hot spots from the patients’ genome in the long-culture *TKO* cells showing aneuploidy, etc.

The reviewer brings up a valid point, particularly since the available RNA-seq data (**Suppl. Fig. 11**) show that *Khdc3* is preferentially expressed in embryonic stem cells and not in somatic cells. To avoid confusion, we have removed the original **Suppl. Fig. 4** from the revised manuscript.

Minor comments:

1. “Deletion” and “depletion” are usually used specifically to regard either DNA level knockout or protein level clearance. The two words were not used properly for multiple times in the manuscript, including title, results and figure legends. Please try to correct them into proper ways: for example, delete *Tet 1/2/3* genes, *Tet1/2/3* deletion, depletion of *TET1/2/3* enzymes.

We thank the reviewer for bringing these errors to our attention and have corrected them in the revised manuscript.

2. In page 3, it is unreasonable to say ‘aneuploidy was not observed in *Tet Tfl* cells not treated with tamoxifen’ given the 20% background aneuploidy in the figure.

NEW Supplementary Figure 5: Bright-field images of *Ctrl* and *Tet TKO* embryos. (A) Phase contrast images (left) from same *Ctrl* and *Tet TKO* embryos shown in Fig. 2F, including DAPI (middle) and γ -H2A.X (right) staining in grayscale. (B) An independent group of embryos was analyzed by confocal microscopy (Airyscan) using Hoechst (left) and γ -H2A.X (right) staining. Yellow arrows and asterisks highlight micronuclei and polar bodies respectively.

We apologize for the misstatement and will change the sentence in the revised manuscript to read “In *Tet triple-floxed* (*Tet Tfl*) mESCs not treated with tamoxifen (“untreated” cells in **Fig. 2C**), the basal frequency of aneuploidy was ~21%.”

3. In figure 4, the volcano plot and the MA-plot are talking about the same thing. Usually people choose one plot type to present.

The two plots provide slightly different information. The volcano plot ranks genes solely by the statistical significance of the fold-change of expression of differentially expressed (DE) genes, and both *Khdc2* and *Khdc3* are highly ranked with *Khdc3* > *Khdc2*. However, this plot does not provide data about the level of gene expression: i.e. it does not necessarily distinguish between different types of 10-fold changes in expression – from 1 to 10, 100 to 1000, or 1000 to 10,000 TPM. The MA-plot provides this important information – its X-axis shows the average expression across the two analyzed samples. However, if the reviewer and editor prefer, we could choose one or another representation to simplify the main figure and move the alternate representation to a supplementary figure.

4. In supp figure 2, why the flag-KHDC3 overexpression has so many bands?

The western blots in the original submission of the manuscript were taken from an initial screen for clones over-expressing *Khdc2* or *Khdc3* and represent unsorted cell populations that contained both mESCs and MEFs. We have now replaced the original blots with **NEW** blots derived from the same sorted ES cell samples that were used for the metaphase spreads (main Figure 5), showing Flag-*Khdc2* and Flag-*Khdc3* protein expression specifically in the transduced ES cells. In our **NEW Suppl. Fig. 10**, we show the same blots for *Ctrl* mESCs together with an immunoblot for Oct4 protein levels, that remain unaltered after re-expression of *Khdc2* or *Khdc3*, since we referred to this point in the original manuscript.

5. As the aneuploidy increases in long period culture of TKO cells, will KHDC3 overexpression still be able to stop further accumulation of aneuploidy or partially rescue the aneuploidy?

As noted by Reviewer #1, aneuploidy does not in fact increase after long-term culture of *Tet iTKO* mESC. To emphasize this point and prevent further misunderstanding, we have included a **NEW** figure panel in **Supplementary Figure 4** showing average levels of aneuploidy from acute and long-term *Tet iTKO* mESC. Since we were interested in early effects of *Tet* genes deletion on chromosome segregation and genome stability, we have not reintroduced *Khdc3* into *Tet iTKO* mESC after prolonged *Tet* gene deletion.

sequencing method, including WGS, Hi-C and WGBS, can be used to detect copy number variations (CNVs) that are present in a substantial fraction of the population and involve large-scale (megabase-level) gains or losses of chromosome arms or whole chromosomes. However, random aneuploidies that are not selected for and so occur only in small fractions of the population will not be detected by bulk sequencing methods at low coverage – single-cell methods such as metaphase spreads or single-cell WGS are needed. This is illustrated by comparing the **NEW Suppl. Fig. 3**, where low coverage WGS on long-term *Tet iTKO* mESC populations (now properly indicated to distinguish them from the acutely deleted samples) did not reveal any evident aneuploidy at the bulk level, while in Figure 2 and Figure 5 using metaphase spreads is clearly revealed a substantial increase in aneuploidy in our *Tet iTKO* mESCs.

Consistently with this, our re-analyses of previously published high coverage WGBS datasets from constitutive *Tet TKO* ESCs models revealed the persistent appearance of aneuploidies in both mouse and human *TET TKO* ESC. Please see details of these analyses in **NEW Suppl. Fig 12**.

6. There is a mistake in using HTseq to count RNA-seq reads. Smart-seq2 protocol can not give the stranded information of the libraries. So “htseq-count -s no” should be used instead of “htseq-count -s yes”.

Thank you for noticing this mistake; we have now corrected it and properly specified that we used “htseq-count -s no”.

7. There is no description in the methods about how the authors evaluated aneuploidy using their WGS data. The labeling of Supp Figure 1 is a little confusing -- why the y-axis is tumor copy number? Is this the copy ratio of a specific set of tumor genes?

Thank you for noticing this mistake; we have changed the Y-axis label of Supplementary Figure 3 to read “Copy Number” instead of “tumor copy number”. We have also revised the Methods section to clarify this point.

8. Just for curiosity—if doing WGS of the long-term-cultured *TKO* cells, is that possible to observe aneuploidy?

If sequencing is performed to sufficient depth, any whole-genome sequencing method, including WGS, Hi-C and WGBS, can be used to detect copy number variations (CNVs) that are present in a substantial fraction of the population and involve large-scale (megabase-level) gains or losses of

Supplementary Figure 12: Disruption of TET and Dnmt genes results in aneuploidies in mESC. Whole Genome Bisulfite Sequencing (WGBS) data from different mESC lines carrying single (A) [74], double or triple (B) [75] *Dnmt* gene knockouts were used to analyze gains (red) and losses (blue) of chromosomal regions compared to their parental *wild-type* mESC cell lines. (C) The *Dnmt TKO* mESC line used in this study (shown in Fig1) also showed aneuploidies. (D) Two replicates from constitutive *Tet TKO* models [64] were also analyzed using high coverage WGBS datasets and compared against their parental lines. (E) human ESC samples lacking *DNMT3A*, *DNMT3B* or both genes [90] were analyzed using similar strategies. Single *DNMT3A* (but not *DNMT3B*) deletion generated a copy number change in chromosome 15 that was also detected in *DNMT3A/B DKO* hESCs. (F) High coverage WGBS samples from constitutive human *TET TKO* ESCs [70] revealed the persistent appearance of aneuploidies also in human *Tet*-deficient cells.

Reviewer #3 (Remarks to the Author):

The effect of *Tet* deficiency on ES cells has not been fully unexplored. It is nice that the authors have taken new effort to characterize the immediate consequence of *Tet* deletion. The observation of aneuploidy in *Tet* TKO ES cells is interesting. Overall, the data presented agree well with the conclusion that downregulation of *Khdc3* might cause the aneuploidy phenotype. Of note, the inclusion of control mESCs in Fig 5B is laudable. However, the manuscript would be improved to be suitable for publication if the issues raised below could be addressed.

We thank the reviewer for these comments and recommendations.

1) (i) Are related chromosomal abnormality observed in *Dnmt* TKO ES cells? (ii) Which of the DNA methyltransferases is responsible for the gain of methylation at *Khdc3* in *Tet* triple knockout ESCs? Upon the deletion of the responsible *Dnmt*, the effect of aneuploidy etc should be reversed. This rescue assay is highly recommended to complement the data shown in Fig 5. *Dnmt* TKO appears to have less severe effect on mESCs and this makes a rescue experiment by *Dnmt* deletion feasible. (iii) Upon the deletion of the responsible *Dnmt*, the effect of aneuploidy etc should be reversed. This rescue assay is highly recommended to complement the data shown in Fig 5. *Dnmt* TKO appears to

have less severe effect on mESCs and this makes a rescue experiment by *Dnmt* deletion feasible.

We agree with the reviewer that these are all interesting questions, and have answered them separately here.

(i) Are related chromosomal abnormality observed in *Dnmt* TKO ES cells?

To ask if aneuploidies were observed in mESC populations lacking *Dnmts*, we analyzed WGBS-datasets for mESC containing single- or multiple deletions of *Dnmt* genes (in which the *Dnmt* genes had been deleted for unspecified periods of time). Indeed, compared with their parental cell lines, mESC with *Dnmt* deletions frequently exhibit aneuploidies (**NEW Suppl. Fig. 12**), a result concordant with previous studies in which genomic instability was noted in cells carrying a hypomorphic *Dnmt1* allele that reduced its expression to 10% of wild-type levels (Gaudet F. *et al.*, 2003 [PMID: 12702876]).

(ii) Which of the DNA methyltransferases is responsible for the gain of methylation at *Khdc3* in *Tet* triple knockout ESCs?

We have addressed this question comprehensively in the **NEW Fig. 4**. To obtain insights into which *Dnmt* family member is responsible for *Khdc3* promoter hypermethylation in WT mESCs, we analyzed ChIP-seq datasets for each of the *Dnmts*

CpGs in the absence of Dnmt3b. *Dnmt3 DKO* showed complete loss of methylation in this region, as did *Dnmt1/3a/b-TKO* mESCs as expected (bottom tracks). These results point to Dnmt3a as the most important de novo Dnmt controlling DNA methylation at the *Khdc3* locus.

We also examined the consequences of sequential deletion of *Tet* and *Dnmt* genes at the *Khdc3* locus (Wang Q et al 2020 [PMID: 32690947]) (NEW Fig. 4D). The investigators first deleted all three *Tet* genes to generate *Tet TKO* mESC; then sequentially deleted *Dnmt3a/b* and *Dnmt1* genes to yield *Tet/Dnmt3 PKO* and *Tet/Dnmt SKO* mESC respectively. WGBS analyses of these cell lines showed that double deletion of *Dnmt3a* and *Dnmt3b* prevented DNA methylation at the *Khdc3* promoter in *Tet TKO* mESC, although unexpectedly, gene body methylation persisted (NEW Figure 4D, compare top three tracks). As expected, the complete *Tet/Dnmt SKO* (sextuple knockout lacking all three Tets and all three Dnmts) completely lacked DNA methylation (NEW Fig. 4D, bottom track). Analysis of RNA-seq datasets showed that the decrease of *Khdc3* expression in *Tet TKO* mESC (NEW Fig. 4E, top and middle tracks) is restored to more than WT levels upon further deletion of both *Dnmt3a* and *Dnmt3b* in *Tet/Dnmt3 PKO* mESC (NEW Figure 4E, bottom track), implying an unexpected crosstalk between Tet and Dnmt proteins.

(Sharif J. et al 2016 [PMID: 27151458]; Baubec T. et al 2015 [PMID: 25607372]), and WGBS datasets from mESC models harboring single *Dnmt3a*, single *Dnmt3b*, double *Dnmt3a/b* (*Dnmt3 DKO*) or triple *Dnmt1/3a/3b* (*DnmtTKO*) gene disruptions compared to their respective controls (WT) (Neri F. et al 2017 [PMID: 28225755]; Haggerty C. et al 2021 [PMID: 34140676]). Dnmt ChIP-seq datasets showed that Dnmt3a is the most enriched Dnmt at regions 5' and 3' of the *Khdc2* and *Khdc3* promoter (NEW Figure 4A, compare top three tracks); both Tet1 and Tet2 are enriched near the transcription start sites (TSS) of both genes, and Tet2 is predominantly present in the gene bodies (NEW Figure 4A, 4th and 5th tracks). These results are consistent with previous reports showing that Dnmt3a plays a major role in controlling DNA methylation in TET-regulated regions in ESC (Zhang X et al 2016; PMID: 27428748), and that Tet1 functions at promoters and Tet2 in gene bodies in mESC (Huang Y. et al PNAS 2014 [PMID: 24474761]). With regard to the control of DNA methylation (NEW Suppl. Fig. 4C), *Dnmt3a* KO mESC (upper tracks) displayed a substantial loss of DNA methylation at the highlighted regions of the *Khdc3* promoter, whereas *Dnmt3b* KO mESC (middle tracks) did not; in fact, certain CpGs in this region showed increased methylation, suggesting relocalization of Dnmt3a or Dnmt1 to these

(iii) Upon the deletion of the responsible *Dnmt*, the effect of aneuploidy etc should be reversed. This rescue assay is highly recommended to complement the data shown in Fig 5. *Dnmt* TKO appears to have less severe effect on mESCs and this makes a rescue experiment by *Dnmt* deletion feasible.

We reanalyzed data from mESC samples carrying deletions of all 3 *Tet* genes and both *Dnmt3* genes (*Penta* KO, PKO) for aneuploidies (Wang Q. et al 2020 [PMID: 32690947]). We found that independently of whether the *Tet* or *Dnmt3* gene deletions were made first (*Tet* → *Dnmt3* PKO versus *Dnmt3* → *Tet* PKO), both trajectories converged in the appearance of aneuploidies. Thus *Dnmt3a/b* deletion does not rescue the aneuploidy phenotype of *Tet*-deficient cells, a result consistent with the fact that single or combined *Dnmt* deletions can themselves result in aneuploidies (Suppl. Fig. 12).

2) Fig. 5, the protein expression levels of lentiviral *Khdc* should be carefully controlled to make sure that they are equivalent of the levels of endogenous proteins in wildtype ESCs. Over-express (OE as indicated in the figure) could lead to an over-rescue thus overlooking other potential mechanisms contributing to the aneuploidy phenotype.

We thank the reviewer for bringing up this point. Although we agree completely, there are still no reliable, commercially available antibodies against *Khdc3*. However, we are now providing detailed panels showing western blots to detect the overexpressed *Khdc2/3*-proteins in our rescue system, including short- and long-exposures (see Figure for reviewers).

3) Transcriptional profiling of *Tet* TKO preimplantation embryos would be important to compare the *in vitro* observations in ESCs with *in vivo* effects. One question is whether *Khdc3* would also be downregulated, concomitantly with increased methylation of CpG dinucleotides in the vicinity of the *Khdc3* gene in the TKO embryos.

We analyzed published datasets from *Tet* TKOs early embryos and found that at E3.5, the *Tet* TKO

blastocysts investigated by Dai HQ. et al (2016) [PMID: 27760115] datasets show significantly lower levels of *Khdc3* gene expression compared to WT. We also analyzed *Tet1/3* DKO blastocyst datasets from Kang J. et al (2015) [PMID: 26199412] and found that *Khdc3* expression is also downregulated in *Tet1/3* DKO blastocytes, albeit to a lesser degree compared to *Tet* TKO blastocysts (NEW Suppl Figure 7C). Thus, *Khdc3* expression is tightly controlled by Tet enzymes both in mESC and *in vivo* in mouse embryos.

NEW Supplementary Figure 7C: The *Khdc3* gene is a conserved target of Tet proteins, that are required to sustain normal expression of *Khdc3* in mESC and in early development. C) RNAs from five Tet1/3 DKO blastocysts (3.5 days) were sequenced and compared to three wildtype (WT) blastocysts. 3 of 5 Tet1/3 DKO blastocysts failed to maintain normal levels of *Khdc3*. (D) Three blastocysts lacking all Tet enzymes (*Tet TKO*) were compared to the same number of control (WT) blastocysts. All *Tet TKO* blastocysts have reduced levels of *Khdc3* compared to their controls.

Please also see our response to point #3 of Reviewer #2.

Other points:

1. Comparison of Tet TKO mESCs that have been maintained in culture, with mESCs of acute Tet deletion in the same experiment would be interesting. How is the change at *Khdc3* in long-term TKO mESCs?

Our qRT-PCR analysis shown in the **NEW Suppl**

Composite from subfigures extracted from Fig 3 and Supp Figure 4: Comparison between the mRNA expression of *Khdc2* and *Khdc3* in our (A) Acute (from Fig3) or (B) Long-term (from Supp Fig4) Tet-deletion systems. Notice that only *Khdc3* (and not *Khdc2*) remain repressed after long-term culture of *Tet iTKO* mESCs.

Fig 4D shows that *Khdc3* mRNA expression is substantially decreased in *Tet iTKO* compared to WT mESCs cultured for prolonged times. To emphasize this point, we prepared a composite figure for the reviewers.

Furthermore, four other previously published studies on *Tet TKO* mESC (Coluccio A. et al 2018 [PMID: 29482634]; Lu F. et

al 2014 [PMID: 25223896]; Reimer M. et al 2019 [PMID: 31286885]; Wang Q. et al 2020 [PMID: 32690947]) confirm the downregulated expression of *Khdc3* relative to WT counterparts (**Suppl. Fig. 7**). In contrast, *Khdc2* mRNA levels in long-term *Tet iTKO* and control mESC are comparable; concomitantly, *Khdc2* mRNA expression varies relative to WT in the published *Tet TKO* systems mentioned above. Our observations confirm that *Khdc3* expression is tightly regulated by TET enzymes in mESCs.

Extracted from Figure 1: **Generation of a tamoxifen-inducible Tet1/2/3 triple-deletion system in mESCs.** ... (B) Genome browser views of *Tet1* and *Tet2* transcripts generated by RNA-Seq analysis of Ctrl (black tracks) and *Tet iTKO* (red tracks) mESC. Note the deletion of the exons uncovering the catalytic domains of the encoded Tet-proteins (purple boxes).

2. It should be explained why 5' exons of *Tet1* and *Tet2* were chosen for deletion rather than 3' exons encoding the catalytic domains. Since the 3' exons are expressed normally (Fig 1b), it is necessary to ensure that no truncated proteins of residual enzymatic activity can be generated in *iTKO* cells. Would the deletion of *Tet1* and *Tet2* exons cause frame-shift mutations?

For both *Tet1* and *Tet2*, we deleted 3' exons (not 5' exons) because these 3' exons encode the core region of the conserved C-terminal catalytic domain. To emphasize this point, we have modified **Fig. 1B** by adding arrows indicating the direction of *Tet1* and *Tet2* gene transcription and a purple bar indicating the exons encoding the catalytic domains of the Tet proteins. We have also mentioned it in the figure legend of **Fig. 1B**.

Supplementary Figure 7: The *Khdc3* gene is a conserved target of Tet proteins, which are required to sustain normal expression of *Khdc3* in mESC and in early development. A, B, Expression of *Khdc3* (A) and *Khdc2* (B) in constitutive Tet triple KO (*Tet TKO*) mESC compared to WT. Left, Genome browser views of the *Khdc3* and *Khdc2* genes taken from four published studies [54, 62-64]. Right, Normalized expression of *Khdc3* and *Khdc2* in each study compared to their corresponding controls (WT). (C) RNAs from five *Tet1/3* DKO blastocysts (3.5 days) were sequenced and compared to three wildtype (WT) blastocysts [38]. 3 of 5 *Tet1/3* DKO blastocysts failed to maintain normal levels of *Khdc3*. (D) Three blastocysts lacking all Tet enzymes (*Tet TKO*) [52] were compared to the same number of control (WT) blastocysts. All *Tet TKO* blastocysts have reduced levels of *Khdc3* compared to their controls.

Supplementary Figure 8: Tet2 maintains DNA demethylation at the *Khdc3* promoter. (A) Genome browser view of the *Khdc3* loci in mESC. The DNA methylation states of WT (red), *Tet1* KO (blue) or *Tet2* KO (black) mESCs were determined by WGBS [89]. (B) The mRNA expression level of *Khdc3* in the same mESC lines was determined by RNA-seq. Note that *Khdc3* is down-regulated in *Tet2* KO but not *Tet1* KO mESC compared to WT.

Our Western blots using *Tet*-iTKO mESCs cell extracts confirmed that no truncated TET proteins were generated; the complete blots will be included in the unprocessed western blots on the final submission. **i)** For the Tet1-antibody (09-872, Millipore-Sigma), on its website, the company said that the immunogen used was: *KLH-conjugated linear peptides corresponding to 12, 14, and 16 amino acids from N-terminal, internal, and C-terminal regions of mouse Methylcytosine dioxygenase TET1*. So, the Tet1-antibody that we used had the capacity to recognize a truncated form, and the fact that we did not detect a lower band means that the mRNA was likely subject to nonsense-mediated decay and no truncated Tet1 protein was produced. **ii)** Similarly, for the Tet2-antibody (ab213369, Abcam), on its website, the company said that the immunogen used was: *Synthetic peptide corresponding to Mouse Tet2 aa 1600-1700 conjugated to keyhole limpet haemocyanin*; this region is encoded in the exon 12, the last and big exon on the figure above, that is not affected by the deletion (in fact it continues being

expressed as RNA). So, the Tet2-antibody also had the capacity to detect a truncated form, and the fact that we did not detect a lower band means that the mRNA was likely subject to nonsense-mediated decay and no truncated Tet2 protein was produced.

3. Which of the Tet enzymes (*Tet1* versus *Tet2*) is more important for maintaining a hypomethylated state for *Khdc3*?

In our new Suppl Fig 8 (included below for examination of Reviewer 3), we analyzed published datasets from Bing Ren's lab (Hon GC, et al 2014 [PMID: 25263596]) and found that single deletion of *Tet2*, but not *Tet1*, resulted in increased CpG methylation within the *Khdc3* promoter region, concomitantly with a reduced expression of *Khdc3* in long-term *Tet2* KO mESCs and not *Tet1* KO mESC. Thus, *Tet2* has a more important role than *Tet1*, both for regulating *Khdc3* expression in mESC and for maintaining the demethylated state of the *Khdc3* promoter. This is consistent with the observation that Tet2 binds

more strongly to the *Khdc3* promoter compared to *Tet1* (Fig. 4F). Please also see our response to Minor Concern 2 of Reviewer 1.

4. *Tet3* is known to be not expressed in mouse ES cells. Could the expression observed in Fig 1B suggest that the cultured *iTKO* ES cells have undergone a certain degree of differentiation?

We thank Reviewer #3 for bringing up this point. It is well known that *Tet3* is very poorly expressed in mESC, and that is true in our mESC cultures as well. The reviewer may have been misled by our Figure 1B, because the scales for the range of y-axis values are very different: 0-1000 for *Tet1* and *Tet2*; 0-60 for *Tet3*. To avoid confusion, we made a new figure with the y-axis scale of the *Tet3* panel to 0-1000 (using the scale of *Tet2* as reference) and this version with the previous one using the autoscale function (with the y-axis scale of 0-60) have moved the *Tet3* panel to **Supp. Fig. 1B** to show the deletion of exon 3 which generates a truncated out-of-frame transcript. In **Supp. Fig. 1A**, we compare the

normalized expression of all *Tet* genes in control mESC.

5. Fig 2, would aneuploidy occur also in *Dnmt TKO* mESCs? This data would be desirable to be included in Fig 2D, if possible. Please see our response to point 1 above. We show now in **NEW Supp. Fig. 12** that deletion of one or more *Dnmts* in mESC is also associated with genomic instability (increased copy number variations and aneuploidies).

REVIEWERS' COMMENTS

Reviewer #1 (Remarks to the Author):

The authors have responded to all of my concerns and I think the revised and improved manuscript is suitable for publication

Reviewer #2 (Remarks to the Author):

All points are properly addressed. The revised manuscript looks very nice.

Reviewer #3 (Remarks to the Author):

My concerns have been addressed.

The new Suppl Fig 9 is a good addition to further demonstrate the role of Khdc3.

Authors may make it clearer that Khdc3 insufficiency cannot account for the genome instability seen in TET2 mutated hematopoietic cells.

FINAL RESPONSE TO REVIEWERS:

REVIEWERS' COMMENTS

Reviewer #1 (Remarks to the Author):

The authors have responded to all of my concerns and I think the revised and improved manuscript is suitable for publication

We thank Reviewer #1 for his/her positive evaluation of our manuscript.

Reviewer #2 (Remarks to the Author):

All points are properly addressed. The revised manuscript looks very nice.

We thank Reviewer #2 for his/her positive evaluation of our manuscript.

Reviewer #3 (Remarks to the Author):

My concerns have been addressed.

The new Suppl Fig 9 is a good addition to further demonstrate the role of Khdc3.

Authors may make it clearer that Khdc3 insufficiency cannot account for the genome instability seen in TET2 mutated hematopoietic cells.

We thank Reviewer #3 for his/her positive evaluation of our manuscript. We followed his/her recommendation about clarifying that *Khdc3*-insufficiency cannot account for the genomic instability seen in TET2 mutated hematopoietic cells by including the next sentence into our Discussion section:

Thus, Khdc3 may operate independently of Khdc2 to prevent chromosome instability in Tet iTKO mESC; alternatively, because Khdc2 mRNA is expressed at higher levels than Khdc3 mRNA in control mESC and is not as markedly downregulated in either acutely-deleted or long-term-cultured Tet iTKO mESC, the two proteins may function as a complex in which KHDC3 protein is limiting whereas KHDC2 protein is not. We note that Khdc3 downregulation cannot account for copy number variations and aneuploidies occurring in hematopoietic malignancies such as DLBCL and PTCL, since Khdc3 expression is restricted to cells at early stages of embryonic development including mESC (Supp. Fig. 11).